# Adaptive Preference Scaling for Reinforcement Learning with Human Feedback

**Ilgee Hong**[*]
Georgia Institute of Technology
ihong39@gatech.edu

**Zichong Li**[*]
Georgia Institute of Technology
zli911@gatech.edu

**Alexander Bukharin**
Georgia Institute of Technology
abukharin3@gatech.edu

**Yixiao Li**
Georgia Institute of Technology
yixiaoli@gatech.edu

**Haoming Jiang**
Amazon
jhaoming@amazon.com

**Tianbao Yang**
Texas A&M University
tianbao-yang@tamu.edu

**Tuo Zhao**
Georgia Institute of Technology
tourzhao@gatech.edu

## Abstract

Reinforcement learning from human feedback (RLHF) is a prevalent approach to align AI systems with human values by learning rewards from human preference data. Due to various reasons, however, such data typically takes the form of rankings over pairs of trajectory segments, which fails to capture the varying strengths of preferences across different pairs. In this paper, we propose a novel adaptive preference loss, underpinned by distributionally robust optimization (DRO), designed to address this uncertainty in preference strength. By incorporating an adaptive scaling parameter into the loss for each pair, our method increases the flexibility of the reward function. Specifically, it assigns small scaling parameters to pairs with ambiguous preferences, leading to more comparable rewards, and large scaling parameters to those with clear preferences for more distinct rewards. Computationally, our proposed loss function is strictly convex and univariate with respect to each scaling parameter, enabling its efficient optimization through a simple second-order algorithm. Our method is versatile and can be readily adapted to various preference optimization frameworks, including direct preference optimization (DPO). Our experiments with robotic control and natural language generation with large language models (LLMs) show that our method not only improves policy performance but also aligns reward function selection more closely with policy optimization, simplifying the hyperparameter tuning process.

## 1 Introduction

In the field of artificial intelligence, aligning AI systems with human preferences has become increasingly crucial, particularly for applications involving complex data and models like large language models (LLMs) in natural language processing [38, 28]. Reinforcement learning from human feedback (RLHF) has gained popularity for customizing AI systems [12, 3, 47]. RLHF involves learning a reward function from human preference data, then using a reinforcement learning algorithm to train a policy to optimize the learned reward model.

---

[*]Equal contribution.

38th Conference on Neural Information Processing Systems (NeurIPS 2024).

A key challenge in RLHF lies in the complexity of reward modeling, which primarily stems from the reliance on preference labels. Since preference labels only provide comparative rankings of trajectory segments without quantifying the scale of underlying preference strengths, previous methods have employed the Bradley-Terry (BT) model [8] in conjunction with cross-entropy loss to learn the reward function from preference data [12, 38]. This approach assumes that the logit of the preference distribution scales linearly with the reward difference across all sample pairs. However, such linear scaling is often insufficient to account for the variations in preference strength among different pairs, restricting the reward function's ability to capture a broader range of reward differences. This restrictive approach to reward modeling limits the flexibility of the learned reward function, hindering its capacity to produce the versatile rewards essential for the downstream policy optimization.

To overcome this shortcoming, we introduce a novel adaptive preference loss function inspired by distributionally robust optimization (DRO) [16]. Our approach incorporates an instance-specific scaling factor to change the scaling between the preference distribution and the reward difference to be non-linear. These factors are learned during training and enable the model to accommodate varying uncertainties of preference strength, thereby enhancing the flexibility of the reward. For pairs showing strong preference (i.e., low preference uncertainty), our method learns a large scaling factor, which enables the model to learn a larger reward difference. In contrast, for pairs showing ambiguous preferences (i.e., high preference uncertainty), our method assigns a smaller scaling factor, enabling the model to learn a smaller reward difference. The additional computational overhead of involving this scaling factor into training is negligible, as the proposed loss function is strictly convex and univariate with respect to each scaling parameter. Therefore, it can be easily optimized by a simple second-order algorithm within a few iterations.

Our experiments on robotic control tasks [39] demonstrate that our method can learn a more flexible reward function, resulting in an improved policy. Surprisingly, we also discover that our method better aligns the learned reward function with downstream policy optimization. Specifically, when tuning hyperparameters for reward modeling, the simplest approach is to select the reward model according to preference prediction accuracy. However, the selected reward function (with the highest accuracy) often yields a downstream policy with poor performance. To address this misalignment, we usually have to jointly tune the parameters across both stages according to downstream policy performance, resulting in significant computational burden and tuning effort. Our proposed method can mitigate this misalignment: When using our adaptive loss, we can select the reward model based on preference prediction accuracy alone and yield a reasonably well-performing policy. This allows separate tuning of the two stages, easing tuning overhead. To our knowledge, the challenge of this misalignment issue is almost untouched in the RLHF literature, and we are the first to propose a principal approach to mitigate this issue.

Moreover, our method is generalizable and can be applied to other preference optimization algorithms. For instance, we implement it with direct preference optimization (DPO) [31] and evaluate its effectiveness on natural language generation tasks using Llama-2 7B [40]. Our results demonstrate that integrating adaptive preference scaling into DPO boosts policy performance, while preserving the benefits of alignment. Alignment is especially critical in this setting, where we employ proprietary models like Claude 3 [1] as judges for policy selection, which demands substantial costs for using the APIs. In the case without access to LLM assessment, we must select policy based solely on preference accuracy, under which our approach substantially outperforms other baselines.

## 2 Related works

**Loss functions for reward learning.** Prior work on this topic is very limited. For example, Song et al. [37] propose using different loss functions for strong and ambiguous preference data in natural language generation tasks. They apply heavy-tailed loss functions for open-ended questions, where preference ambiguity is desirable, and light-tailed loss functions for close-ended questions requiring clear-cut rewards. However, their approach requires knowing the question type a priori, necessitating extra labeling effort, and may fail for complex questions containing both open and closed aspects. Zhao et al. [47] propose using a hinge loss, which results in zero gradient when the learned reward difference exceeds a margin of 1. This limits the ability to learn very large differences in rewards. Azar et al. [2] develop $\Psi$ Preference Optimization with Identity Mapping (IPO), which modifies DPO with a loss function matching the scaling of KL-divergence between the learned policy and the initial policy to avoid overfitting due to weak regularization. In contrast to prior work, our method is

more broadly applicable to complex preference learning tasks without needing additional labeling or sacrificing the ability to learn arbitrarily large reward differences.

**Adaptive temperature scaling (ATS).** Temperature scaling (TS) aims to adjust the entropy of probabilistic models by rescaling their logit outputs before the softmax function is applied. This simple method not only enables confidence calibration [19], but also plays a vital role in various machine learning methods, including knowledge distillation [20], reinforcement learning [25], and contrastive learning [43]. Building on TS, adaptive temperature scaling (ATS) enhances flexibility by using instance-specific scalars. Most ATS method trains an additional network for predicting the temperature parameter, which is further integrated into the softmax operator to calibrate the prediction probabilities [44, 14, 4, 21]. In contrast to the aforementioned ATS methods, the proposed adaptive preference scaling (APS) is not designed for classical confidence calibration, but is crafted specifically to enhance the training process of reward function in RLHF. Consequently, the interpretations of scaling factors in ATS and APS are *opposite*. In ATS, a larger scaling parameter is applied to data with higher uncertainty (e.g., data that the classifier is likely to misclassify), which reduces the magnitude of the corresponding logit. Conversely, in APS, a larger scaling factor is assigned to data with clearer preferences, resulting in a larger logit. This distinction clarifies why the scaling parameter in our approach does not correspond to the concept of "temperature" from statistical physics. Additionally, we propose a principled framework for learning scaling parameter based on DRO, which avoids the complexities of designing specific temperature networks and does not rely on heuristically designed loss functions.

**Distributionally robust optimization (DRO).** DRO is a technique that trains machine learning models to be robust against uncertainty in the data distribution. Specifically, DRO finds a solution that performs well under the worst-case distribution within a specified uncertainty set around the empirical data distribution [5, 7, 23, 35, 16]. DRO has been applied in various AI/ML domains to improve generalization when the test distribution differs from the training distribution [27, 18, 26, 10, 45, 30]. Our framework is motivated by Qi et al. [30], which tackles KL-constrained DRO problem. However, our approach differs in two significant ways. First, instead of using a single KL constraint for the entire training dataset, we apply a separate KL constraint to each individual training data. Second, since each training data involves just two distributional variables, we can use a deterministic method to optimize these efficiently. Note that while our proposed method is inspired by DRO, it serves a distinct purpose: improving reward learning in RLHF, which is *orthogonal* to distributional robustness.

## 3 Method

In this section, we first outline the problem setup, derive the loss function with adaptive preference scaling, and then provide theoretical motivation for our proposed loss. At last, we present an optimization algorithm, extend the approach to direct preference optimization, and introduce a variant of our proposed loss that incorporates quadratic regularization.

### 3.1 Problem setup

We consider a reward-free Markov decision process $\mathcal{M} = (\mathcal{S}, \mathcal{A}, p, \gamma)$ with state $s \in \mathcal{S}$, action $a \in \mathcal{A}$, state transition function $p$, and discount factor $\gamma$. The ground truth reward function $r : \mathcal{S} \times \mathcal{A} \to \mathbb{R}$ is assumed to be unknown, but only human preferences over pairs of trajectory segments are observed. A trajectory segment is a sequence of consecutive state and action pairs $z = \{(s_m, a_m), (s_{m+1}, a_{m+1}), \ldots, (s_{k-1}, a_{k-1})\} \in (\mathcal{S} \times \mathcal{A})^{k-m}$. We denote $z_1 \succ z_2$ to indicate that the human preferred trajectory segment $z_1$ over the trajectory segment $z_2$ and denote the preferred one with a subscript $w$ and the dispreferred one with a subscript $l$ (i.e., $z_w$ and $z_l$). Here, we are given a human preference dataset of trajectory segments $\mathcal{D}_{\text{pref}} = (z_{w,i}, z_{l,i})_{i=1}^N$. Our goal is to find a reward $\hat{r}(s, a)$, which is well-aligned with human preferences. Once we learn the reward, we then find a policy $\pi \in \Delta_{\mathcal{A}}^{\mathcal{S}}$ such that it maximizes the expected sum of discounted rewards,

$$\max_\pi \mathbb{E}_{z \sim \mathcal{D}_\pi} \big[ \hat{r}(z) \big],$$

where $\hat{r}(z) = \sum_{(s_t, a_t) \in z} \gamma^t \hat{r}(s_t, a_t)$ and $\mathcal{D}_\pi$ denotes the stationary distribution of the state-action pair induced by $\pi$.

## 3.2 Reward learning with adaptive preference scaling

We now focus on the reward learning phase in RLHF, a crucial stage for capturing human preferences across various trajectory segments. The standard reward learning procedure assumes that the reward function determines a preference distribution, also known as the Bradley-Terry (BT) model [8],

$$p_r(z_w \succ z_l) = \sigma(r(z_w) - r(z_l)), \tag{1}$$

where $\sigma$ denotes the sigmoid function. The reward function is then learned by minimizing the expectation of negative log-likelihood of $r$ over the preference data [12]:

$$\min_r \ \mathcal{L}_{\mathrm{pref}}(r) = -\mathbb{E}_{(z_w, z_l) \sim \mathcal{D}_{\mathrm{pref}}} \big[ \log p_r(z_w \succ z_l) \big]. \tag{2}$$

As can be seen from (1), the BT model essentially assumes that the logit of the preference distribution $\sigma^{-1}(p_r(z_w \succ z_l))$ scales linearly with the reward difference, regardless of the specific pair of samples. Such linear scaling, however, may not align well with downstream policy learning. Human preferences are often influenced by numerous factors that interact in non-linear ways, making the BT model suboptimal as a reward model. For example, when the reward difference is small, even slight changes in certain features might lead to significant shifts in preference. The BT model may struggle to capture such rapid shifts due to its slower transition.

To address this challenge, we propose an adaptive preference loss based on KL-constrained distributionally robust optimization formulation [30], which can implicitly change the scaling between the logit and the reward difference to be non-linear. Specifically, given a pair of trajectory segments $(z_1, z_2)$, we denote $d_r(z_1, z_2) = \mathbf{1}(z_1 \succ z_2) \cdot (r(z_2) - r(z_1))$ and $p = (p_1, p_2)$. We define the following instance-level loss:

$$\ell_r(z_1, z_2) := \max_{p \in \Delta_2} p_1 d_r(z_1, z_2) + p_2 d_r(z_2, z_1) - \tau_0 \mathrm{KL}(p, 1/2) \qquad \text{s.t.} \quad \mathrm{KL}(p, 1/2) \leq \rho_0, \tag{3}$$

where $\Delta_2 = \{p \in \mathbb{R}^2 : p_1 + p_2 = 1, 0 \leq p_1, p_2 \leq 1\}$, $1/2$ is denoted for the uniform distribution, and $\rho_0, \tau_0 > 0$ are shared prespecified parameters across all instances. $\mathrm{KL}(\cdot, \cdot)$ denotes the KL divergence. Note that without the KL-constraint, (3) is reduced to the cross-entropy loss with $\tau_0 = 1$. Unlike general KL-constrained DRO formulation, which considers a distribution $p$ over all training samples, the distributional variable $p$ in (3) is associated specifically with binary preference comparisons for each pair.

We then convert (3) into an equivalent minimax formulation based on the Lagrangian duality,

$$\min_{\lambda \geq 0} \max_{p \in \Delta_2} p_1 d_r(z_1, z_2) + p_2 d_r(z_2, z_1) - (\lambda + \tau_0)(\mathrm{KL}(p, 1/2) - \rho_0),$$

where $\lambda$ is the Lagrange multiplier. By defining $\tau$ as $\tau = \lambda + \tau_0$ and applying the optimality condition for $p$, we have

$$\min_{\tau \geq \tau_0} \ -\tau \log p_{r,\tau}(z_w \succ z_l) + (\rho_0 - \log 2)\tau, \tag{4}$$

where

$$p_{r,\tau}(z_w \succ z_l) = \sigma\left(\frac{r(z_w) - r(z_l)}{\tau}\right). \tag{5}$$

We refer to Appendix A.1 for the full derivation. Note that the preference scaling factor $\tau$ in (4) and (5) serves as the Lagrange multiplier of (3). This scaling parameter $\tau$ is used specifically for training the reward function $r$, rather than calibrating the preference distribution $p_{r,\tau}(z_w \succ z_l)$. The scaler $\tau$ is used exclusively during the reward learning phase and is no longer needed in subsequent policy optimization, where the reward function $r$ alone is used.

Moreover, the scaling parameter $\tau$ is defined to be an instance-specific parameter corresponding to the pair of trajectory segments $(z_w, z_l)$. Therefore, when applying our adaptive loss to reward learning, for each pair $(z_{w,i}, z_{l,i})$, we need to define a corresponding scaling parameter denoted by $\tau_i$. The overall loss function over the training set $\mathcal{D}_{\mathrm{pref}}$ is as follows:

$$\min_{r, \tau_1, \ldots, \tau_N \in \Omega} \frac{1}{N} \sum_{i=1}^N \ell_i(r, \tau_i) := \frac{1}{N} \sum_{i=1}^N \big( -\tau_i \log p_{r,\tau_i}(z_{w,i} \succ z_{l,i}) + \rho \tau_i \big), \tag{6}$$

where $T = (\tau_1, \ldots, \tau_N)$, $\Omega = \{\tau : \tau_0 \leq \tau \leq \tau_{\max}\}$ with $\tau_{\max}$ as another prespecified parameter, and $\rho = \rho_0 - \log 2 > -\log 2$. Here, we also involve an upper bound $\tau_{\max} > 0$ in (6), and we will explain why it is needed in the next subsection.

### 3.3 Theoretical insights

We next provide some theoretical insights on why the scaling parameter $\tau$ can help gain adaptivity by a proposition. For simplicity, we only consider a pair of trajectories.

**Proposition 3.1.** *Assume we have a pair of trajectories $z_1, z_2$, and the preference distribution $p(z_1 \succ z_2) = p^* \in (0, 1)$, i.e., the probability, that $z_1$ is preferred over $z_2$, is $p^*$. Consider the problem of minimizing the expectation of our adaptive loss function over the preference distribution:*

$$\min_{r, \tau \in \Omega} -\tau p^* \log\left(\sigma\left((r(z_1) - r(z_2))/\tau\right)\right) - \tau(1 - p^*) \log\left(\sigma\left((r(z_2) - r(z_1))/\tau\right)\right) + \rho\tau. \quad (7)$$

*Then the minimizer $\tau^*$ and $r^*$ of the expected loss satisfy*

$$\tau^* = \begin{cases} \tau_0 & \text{if } -p^* \log(p^*) - (1 - p^*) \log(1 - p^*) + \rho > 0, \\ \tau_{\max} & \text{if } -p^* \log(p^*) - (1 - p^*) \log(1 - p^*) + \rho < 0, \end{cases}$$

$$r^*(z_1) - r^*(z_2) = \tau^* \sigma^{-1}(p^*).$$

*Here, $\sigma^{-1}$ is the inverse of sigmoid function.*

Note that the expected loss (7) is only for easing theoretical analysis, as $p^*$ is not accessible in practice. From Proposition 3.1, we can see that when $p^*$ is close enough to $0.5$, (i.e., the uncertainty of preference is large), the corresponding optimal $\tau^*$ is at the lower bound $\tau_0$. The resulting optimal reward difference is $\tau_0 \sigma^{-1}(p^*)$, which is smaller than the counterpart obtained by the cross-entropy loss when $\tau_0 < 1$. Conversely, when $p^*$ is close to $0$ or $1$, (i.e., the uncertainty of preference is small), the resulting optimal $\tau^*$ is at the upper bound $\tau_{\max}$. Here, we introduce the upper bound $\tau_{\max}$ to ensure that the optimal $\tau^*$ is bounded. The resulting reward difference in this case is $\tau_{\max} \sigma^{-1}(p^*)$, which is larger than the counterpart obtained by the cross-entropy loss when $\tau_{\max} > 1$. Our theoretical analysis suggests that the adaptive scaling factor essentially changes the correspondence between the logit of preference distribution and the reward difference for each pair of trajectory segments, which could lead to a more flexible reward model.

We further visualize our adaptive preference loss in Figure 1, setting $\tau_0$ and $\tau_{\max}$ to $0.1$ and $5.0$, respectively. As depicted, our adaptive preference loss behaves distinctly compared to the cross-entropy loss. With large learned reward differences, the cross-entropy tends to be very flat, while our loss maintains a non-trivial gradient, allowing us to continually decrease the loss function. In contrast, for small positive learned reward differences, our loss yields a smaller gradient, thereby less encouraging the reward model to further distinguish pairs of ambiguous trajectory segments. This is consistent with our theoretical analysis.

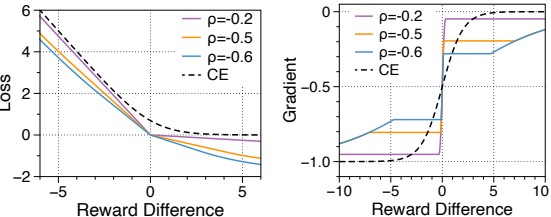

Figure 1: Visualization of the loss function (left) and its gradient (right) on different reward differences.

### 3.4 Algorithm

We present an efficient algorithm for solving (6). Suppose we parameterize $r$ as a neural network with parameter $\phi$. At the $m$-th iteration, we have the iterate $\phi^{(m)}$, and we sample a pair of trajectory segments $z_{w,i}$ and $z_{l,i}$. We initialize $\tau_i^{(0)} = 1$ and then optimize $\tau_i$ by a projected Newton method subject to a simple interval constraint $\Omega$ [6]. Specifically, for $k = 0, ..., K - 1$, we take

$$\tau_i^{(k+1)} = \underset{\tau_i \in \Omega}{\Pi} (\tau_i^{(k)} + \Delta_i^{(k)}), \quad (8)$$

where $\Delta_i^{(k)}$ denotes the descent direction

$$\Delta_i^{(k)} = -\frac{\nabla_{\tau_i} \ell_i(\phi^{(m)}, \tau_i^{(k)})}{\nabla^2_{\tau_i} \ell_i(\phi^{(m)}, \tau_i^{(k)})}. \quad (9)$$

Once we get $\tau_i^{(K)}$, we update $\phi$ by a stochastic gradient descent step

$$\phi^{(m+1)} = \phi^{(m)} - \eta_\phi \nabla_\phi \ell_i(\phi^{(m)}, \tau_i^{(K)}), \quad (10)$$

where $\eta_\phi$ is the learning rate. We summarized our proposed algorithm in Algorithm 1, which is presented in a per-data manner for clarity but can be directly adapted for mini-batch learning.

---

**Algorithm 1** Algorithm for reward learning with adaptive preference scaling

---
1: **Input**: $\tau_0, \tau_{\max}, \rho, \eta_\phi$;
2: **for** $m = 0, 1, 2, \ldots, M - 1$ **do**
3:      Sample a pair of trajectory segments from $\mathcal{D}_{\mathrm{pref}}$;
4:      Set $\tau_i^0 = 1$;
5:      **for** $k = 0, 1, 2, \ldots, K - 1$ **do**
6:          Compute $\Delta_i^{(k)}$ using (9) and update $\tau_i^{(k)}$ using (8);
7:      **end for**
8:      Update $\phi^{(m)}$ using (10) or Adam-style step;
9: **end for**

---

**Remark 3.1.** Note that since $\ell_i(\phi, \tau_i)$ is strictly convex and univariate with respect to $\tau_i$, in each iteration $m$, $\tau_i^{(K)}$ is guaranteed to be near-optimal (i.e., $\tau_i^{(K)} \approx \tau_i^\star$). Therefore, the convergence of Algorithm 1 can be guaranteed by the convergence of stochastic gradient descent on the reward model parameter $\phi$.

**Remark 3.2.** Computationally, Algorithm 1 incurs negligible additional cost. The inner minimization problem (Lines 5-7 in Algorithm 1) can be solved to near-optimality efficiently within a few iterations (e.g. $K = 5$) given its convex and univariate nature. The additional overhead of each update is minor compared to the overall RLHF pipeline.

### 3.5 Extension to direct preference optimization (DPO)

Our adaptive preference scaling approach is generic and can be extended to DPO [31], which is another popular method for policy learning from human preferences. DPO directly learns the policy in supervised manner using the preference data of state-action pairs $\mathcal{D}_{\mathrm{pref}} = (s_i, a_{w,i}, a_{l,i})_{i=1}^N$. This approach forgoes the need to learn the reward function explicitly by the reparameterization of reward function $r$ with respect to its optimal policy $\pi_r$,

$$r(s, a) = \beta \log(\pi_r(a|s)/\pi_{\mathrm{ref}}(a|s)) + \beta \log Z(s), \tag{11}$$

where $Z(s) = \sum_a \pi_{\mathrm{ref}}(a|s) \exp\left(r(s, a)/\beta\right)$ and $\pi_{\mathrm{ref}}$ denotes the reference policy. By plugging in (11) back into (2), we have the policy optimization problem

$$\min_\pi \mathcal{L}_{\mathrm{DPO}}(\pi) = -\mathbb{E}_{(s, a_w, a_l) \sim \mathcal{D}_{\mathrm{pref}}} \log \sigma\left(\beta r_\pi(a_w|s) - \beta r_\pi(a_l|s)\right),$$

where $r_\pi(a|s) = \log(\pi(a|s)/\pi_{\mathrm{ref}}(a|s))$ denotes the log-probability ratio.

Similarly. we can integrate adaptive preference scaling into DPO by plugging in (11) into (6). By merging $\beta$ with the $\tau_i$ and $\rho$, we can further obtain the adaptive DPO (Ada-DPO) formulation as

$$\min_{\pi, \tau_1, \ldots, \tau_N \in \Omega} \mathcal{L}_{\mathrm{Ada-DPO}}(\pi, \tau_1, ..., \tau_N) := \frac{1}{N} \sum_{i=1}^N \left[ -\tau_i \log \sigma\left( \frac{r_\pi(a_{w,i}|s_i) - r_\pi(a_{l,i}|s_i)}{\tau_i} \right) + \rho \tau_i \right].$$

**Remark 3.3.** Note that the proposed adaptive preference loss can be further combined with other RLHF approaches, such as PEBBLE [24], SURF [29], and PARL [11], which still optimize the standard cross-entropy loss (see [24, Eq. (4)], [29, Eq. (3)], and [11, Eq. (5)]).

### 3.6 Extension to quadratic regularization

We now introduce a variant of our adaptive preference loss that uses quadratic regularization for $\tau$. This modification removes the need for the hyperparameter $\tau_{\max}$ in $\Omega$, easing the tuning effort. We define the following instance-level adaptive preference loss with quadratic regularization:

$$\min_{\tau \geq \tau_0} \ell_{\mathrm{quad}}(r, \tau) := -\tau \log p_{r,\tau}(z_w \succ z_l) + \rho_0 \tau^2 - \log 2\tau. \tag{12}$$

Compared to (4), which includes a linear regularization term of $(\rho_0 - \log 2)\tau$, (12) modifies the regularization term with coefficient $\rho_0$ to be quadratic while keeping the term $\log 2\tau$ linear. Additionally, in (12), the constraint on $\tau$ only specifies a lower bound $\tau_0$ and no longer includes an upper bound $\tau_{\max}$. The following proposition provides theoretical insights for this modification.

**Proposition 3.2.** *Assume we have a pair of trajectories $z_1, z_2$, and the preference distribution $p(z_1 \succ z_2) = p^* \in (0, 1)$. Consider the problem of minimizing the expectation of our adaptive loss function with quadratic regularization over the preference distribution:*

$$\min_{r, \tau \geq \tau_0} -\tau p^* \log \left( \sigma \left( (r(z_1) - r(z_2))/\tau \right) \right) - \tau (1 - p^*) \log \left( \sigma \left( (r(z_2) - r(z_1))/\tau \right) \right) + \rho_0 \tau^2 - \log 2\tau.$$

*Then the minimizer $\tau^*$ and $r^*$ of the expected loss satisfy*

$$\tau^\star = \max\{\tau_0, (p^* \log(p^*) + (1 - p^*) \log(1 - p^*) + \log 2)/(2\rho_0)\},$$

$$r^*(z_1) - r^*(z_2) = \tau^* \sigma^{-1}(p^*).$$

*Here, $\sigma^{-1}$ is the inverse of sigmoid function.*

Note that unlike the adaptive preference loss with linear regularization described in Proposition 3.1, the optimal value $\tau^\star$ for quadratic regularization does not involve the upper bound $\tau_{\max}$.

## 4 Experiments

In this section, we examine the effectiveness of our adaptive preference loss based on robotic control and natural language generation tasks. Due to space limit, we defer the experiments with quadratic regularization, ablation studies, and discussions on hyperparameter selection to Appendix C.

### 4.1 Robotic control

**Experiment setup.** We apply our proposed reward learning method on 3 robotic control tasks from the PyBullet [13] environments: *HalfCheetah*, *Ant*, and *Hopper*. These environments are similar to those available in OpenAI Gym [9] but they are known to be much harder to solve [34]. Similarly to Gao et al. [17], our setting is synthetic, where we use the ground truth rewards to provide preference labels on each pair of samples due to high expense of collecting human preferences. For the reward function, we use two-hidden-layer MLPs, each containing 64 hidden units. This configuration is aligned with the designs of both the policy and value networks. Following Christiano et al. [12], we repeat the following three steps for each stage: (i) We sample a set of trajectories by the policy $\pi$, and update the policy with proximal policy optimization (PPO, Schulman et al. [36]) alongside a reward function $\hat{r}$. (ii) We split the segments (the sequence of state-action pairs) into a training set and a testing set. Then, we randomly sample pairs of segments from the training set, and generate $\mathcal{D}_{\text{pref}}$ with preference labels. We do the same to the testing set, and generate $\mathcal{D}'_{\text{pref}}$. (iii) We train the reward function $\hat{r}$ on $\mathcal{D}_{\text{pref}}$, and use $\mathcal{D}'_{\text{pref}}$ for evaluating the preference prediction of $\hat{r}$.

For notational simplicity, we name our proposed adaptive preference scaling method for reward learning as "Ada-Pref". We compare Ada-Pref with the baseline method "Pref", which uses the standard cross-entropy loss for reward learning. For every 10000 timesteps the policy $\pi$ runs, we evaluate the learned policy based on 20 test episodes. We also compute the average preference prediction accuracy of the learned reward function across stages. We set the budget to 3 million timesteps and perform training over 10 different seeds. For hyperparameter tuning in both reward learning and policy optimization, we apply two different criteria: 1) We identify the best policy based on its performance (the one with the highest return) and subsequently select the corresponding reward function. 2) We choose the best reward function based on its performance (the one with the highest average preference prediction accuracy) and then select the corresponding policy. Details of the implementations and hyperparameter tuning procedures are in Appendix B.1.

**Results.** We summarize the results on three PyBullet tasks as follows:

Table 1 and Figure 2 illustrate the results for Pref and Ada-Pref on the PyBullet tasks, based on the first hyperparameter tuning criterion. In Table 1, we report the highest return of the best policy and the average preference accuracy of the corresponding reward function. We can see that Ada-Pref consistently outperforms Pref in terms of return on all three tasks and achieves comparable preference accuracy. The upper panel of Figure 2 shows the learning curve plots. We can see that Ada-Pref surpasses Pref at nearly every timestep and reaches a higher plateau across all tasks. The lower panel of Figure 2 presents percentile plots from different seeds to demonstrate individual run behaviors. As shown, we confirm that Ada-Pref consistently outperforms Pref at every percentile across all tasks.

Table 2 presents the results for Pref and Ada-Pref based on the second hyperparameter tuning criterion. From Table 2, we can see that both methods show a decrease in performance compared to Table 1,

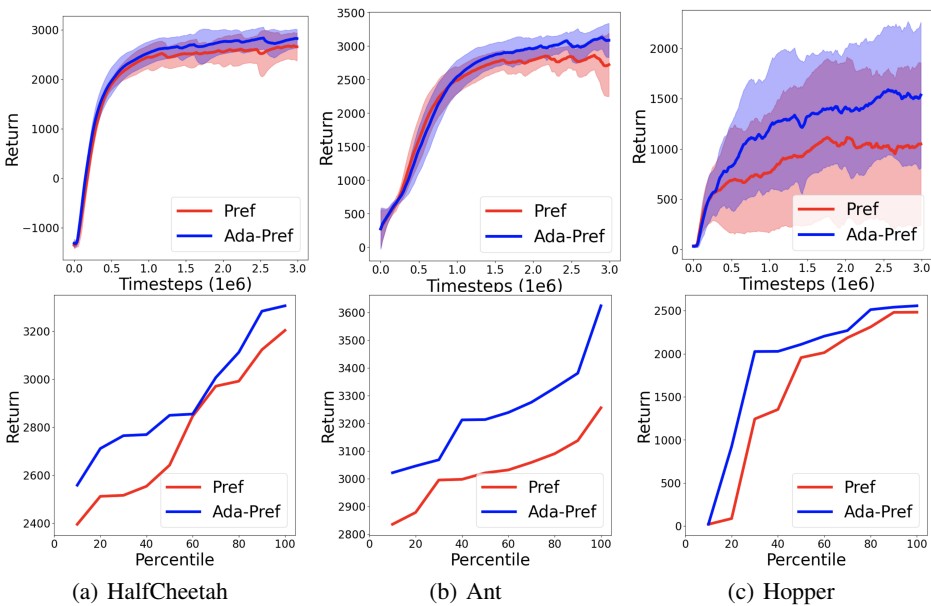

Figure 2: Learning curve plots (top) and percentile plots (bottom) for Pref and Ada-Pref. For the learning curve plots, returns at each timestep are averaged across 10 different seeds, then smoothed over timesteps using an exponential moving average (EMA) with a smoothing factor of $\alpha = 0.1$. For the percentile plots, returns from 10 different seeds are sorted in ascending order.

while Ada-Pref still outperforms Pref in terms of both preference accuracy and return on all three tasks. Furthermore, Ada-Pref demonstrates greater resistance to performance degradation than Pref, indicating its superior ability to align the learned reward function with policy optimization. This alignment allows for effective policy selection based on preference accuracy without the need to evaluate the policy using ground truth rewards.

Table 1: Table for the highest return of the best policy and the average preference prediction accuracy of the corresponding reward function.

| Task | Method | Return | Preference Accuracy (%) |
|---|---|---|---|
| HalfCheetah | Pref | 2724.42 | 89.09 |
| | Ada-Pref | **2875.45** | 89.46 |
| Ant | Pref | 2917.81 | 85.57 |
| | Ada-Pref | **3177.11** | 85.48 |
| Hopper | Pref | 1324.91 | 92.08 |
| | Ada-Pref | **1692.10** | 91.36 |

Table 2: Table for the average preference prediction accuracy of the best reward function and the highest return of the corresponding policy.

| Task | Method | Return | Preference Accuracy (%) |
|---|---|---|---|
| HalfCheetah | Pref | 2620.83 | 89.41 |
| | Ada-Pref | **2865.07** | 90.75 |
| Ant | Pref | 2750.99 | 87.93 |
| | Ada-Pref | **3008.69** | 89.23 |
| Hopper | Pref | 744.66 | 93.18 |
| | Ada-Pref | **1134.73** | 93.26 |

## 4.2 Natural language generation

**Experiment setup.** We apply DPO with our proposed adaptive loss (Ada-DPO) method to two open-ended text generation tasks: *summarization* and *single-turn dialogue*. We adopt the Llama-2 7B model [40] as the backbone and conduct instruction tuning on each task to obtain the initial reference models. For summarization, the policy generates summaries given posts collected from Reddit. We use the filtered TL;DR summarization dataset [41] for instruction tuning, which contains more than 117K Reddit posts, each with a human-written summary. We apply the human preferences collected by Stiennon et al. [38] for preference optimization, where each transcript contains a pair of responses along with a preference label. For single-turn dialogue, the policy responds to various human queries ranging from simple questions to complex demands. We utilize the Anthropic Helpful and Harmless dialogue preferences dataset [3] for both instruction tuning and preference optimization. This dataset contains 170k human-AI dialogues, with each dialogue containing two AI responses and a human preference label. We use the preferred responses for instruction tuning and the full set of preferences for optimization. For instruction tuning stage, we fine-tune the entire Llama-2 model. For the alignment stage using Ada-DPO and different baselines, we apply LoRA fine-tuning for computational efficiency concerns, as we need to simultaneously tune multiple hyperparameters.

The rank of the LoRA adaptor is 64. We consider three baseline methods: DPO [31], Ψ Preference Optimization with Identity Mapping (IPO) [2] and Sequence Likelihood Calibration with Human Feedback (SLiC-HF) [47].

As human evaluation is prohibitively expensive, we use Claude 3 [1], a proprietary large language model, to automatically evaluate responses based on summary quality and helpfulness/harmlessness for the summarization and dialogue tasks, respectively. Prior work has shown that Claude 3 and GPT-4 can effectively measure a quantitative improvement over the instruction-tuned model [15]. We split a small subset from each instruction tuning dataset for testing and calculate the win rate against the instruction-tuned reference model as the evaluation metric. The percentage of instances where the response generated by policy A is preferred over policy B is referred to as the win rate of A against B. We also split a subset from each preference optimization dataset to validate the preference prediction accuracy. Details of the implementations and hyperparameter selections are in Appendix B.2.

**Results.** We summarize the results on the two natural language generation tasks as follows:

In Figure 3, we select the model with the highest win rate and present the win rate and its preference accuracy for all baselines. We observe that Ada-DPO outperforms the other baselines on both tasks in terms of win rate and achieves comparable preference accuracy. In Figure 4, we display the performance of the model selected with the highest accuracy (not win rate). As shown, Ada-DPO achieves a significant improvement beyond the DPO baseline in terms of win rate and obtains a comparable preference accuracy. This again indicates that Ada-DPO yields better alignment between the learned reward function and policy optimization, allowing good policy selection based on preference accuracy without a proprietary LLM judge.

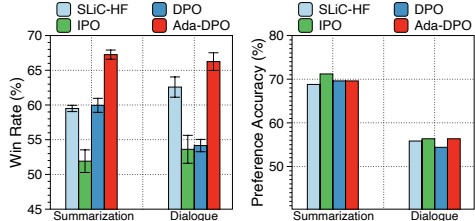

Figure 3: The best win rate and the preference prediction accuracy of the corresponding model.

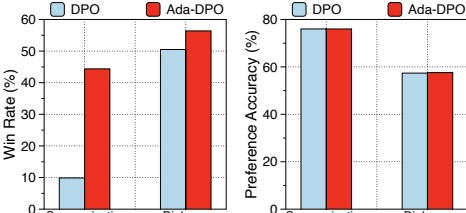

Figure 4: The best preference prediction accuracy and the win rate of the corresponding model.

### 4.3 Detailed analysis

We present detailed analyses of Ada-Pref and Ada-DPO for both the Ant and summarization tasks. Figure 5(a) presents a histogram of the learned scaling factors $\tau$ for the Ant task. We can see that around 60% of these scaling factors reach the upper bound, while about 10% converge to the lower bound, and the rest are distributed across the region. In Figure 5(b), we explore the relationship between preference strength and the learned scaling factors $\tau$, and in Figure 5(c), we investigate the relationship between preference strength and the learned reward difference for Pref and Ada-Pref. We measure preference strength using the true reward difference, categorize it into five percentile bins, and then bin the scaling factors and the learned reward differences accordingly to compute the average. As can be seen, the learned scaling factor increases monotonically with preference strength, demonstrating that the our method successfully adapts the loss scaling to the varying degrees of preference in the data. Furthermore, Ada-Pref learns smaller reward differences for pairs with ambiguous preferences and learns larger reward differences for those with strong preferences, compared to Pref. This indicates that our method leads to a more flexible reward function.

In Figure 6(a), we plot a histogram of the learned scaling factors $\tau$ for the summarization task. We can see that around 40% of the scaling factors converge to the upper bound, with the rest distributed across the region. We also display the relationship between the confidence scores and the scaling factors in Figure 6(b). The confidence score is an integer from 1 to 4 included in the dataset, and a higher score denotes a stronger preference. We bin the scaling factors based on confidence scores and compute the average. As shown, the scaling factors positively correlate with confidence scores, justifying that we learn larger $\tau$ for strong preferences and smaller $\tau$ for ambiguous ones.

We further present two pairs of preference samples where Ada-DPO assigns large or small scaling factors in Figure 7. We observe that the sample pair with a large scaling factor shows a strong

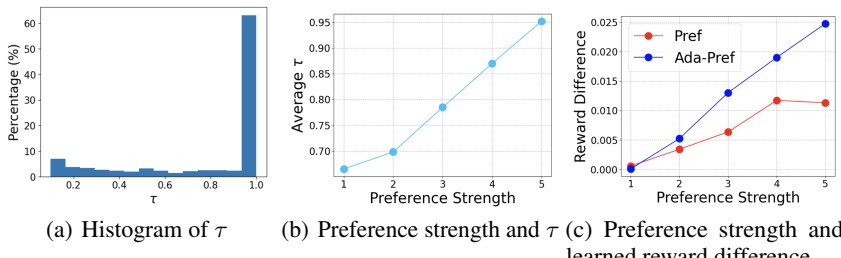

(a) Histogram of $\tau$     (b) Preference strength and $\tau$     (c) Preference strength and learned reward difference

Figure 5: Histogram of learned scaling factors, relationship between preference strength and the learned scaling factors, and relationship between preference strength and the learned reward difference. All plots are from the Ant task.

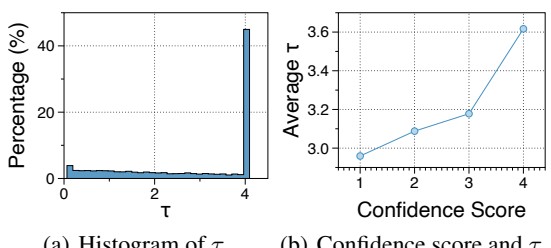

(a) Histogram of $\tau$     (b) Confidence score and $\tau$

Figure 6: Histogram of learned scaling factors and relationship between the confidence scores and the learned scaling factors. Both plots are from the summarization task.

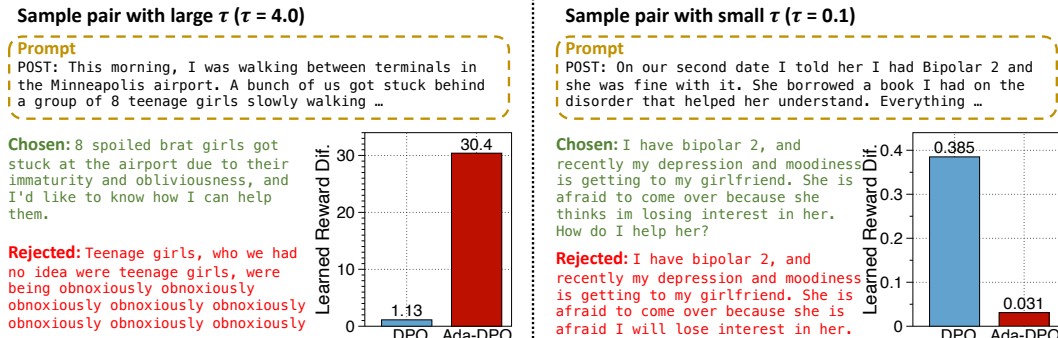

Figure 7: Examples of preference sample pairs with large (left) and small (right) scaling factors $\tau$, and the comparison of the learned reward difference. The preferred (chosen) responses are colored by green and the rejected responses are colored by red.

preference, as the rejected response is nonsensical while the chosen one is clear. Ada-DPO learns a larger reward difference for such data, while it is much smaller with DPO. Conversely, for the sample pair with a small scaling factor, the two responses are very similar, indicating its ambiguity. Ada-DPO learns a small reward difference on this pair, while DPO gets a large reward difference.

# 5 Conclusion

RLHF is an emerging challenge in machine learning. Prior to the popularity of models like ChatGPT, research on designing proper loss functions for reward learning was limited. To bridge this gap, we explore uncertainties in underlying preference strengths and propose an adaptive preference loss function. This loss function incorporates instance-specific scaling factors to modulate the correspondence between reward differences and preference distributions. Taking the result in this paper as an initial start, we expect more sophisticated and stronger follow-up work that applies to RLHF with similar structures. All of these efforts may ultimately assist in developing more principled RLHF methods to better control risks associated with advanced AI systems.

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

# A  Derivation and proofs of Section 3

## A.1  Derivation of Equation (4)

In this subsection, we present the full derivation of Equation (4). Recall the following loss:
$$\ell_r(z_1, z_2) = \max_{p \in \Delta_2} p_1 d_r(z_1, z_2) + p_2 d_r(z_2, z_1) - \tau_0 \mathrm{KL}(p, 1/2) \qquad \text{s.t.} \quad \mathrm{KL}(p, 1/2) \leq \rho_0.$$
Using the Lagrangian duality, we have
$$\max_{p \in \Delta_2} \min_{\lambda \geq 0} p_1 d_r(z_1, z_2) + p_2 d_r(z_2, z_1) - \tau_0 \mathrm{KL}(p, 1/2) - \lambda(\mathrm{KL}(p, 1/2) - \rho_0).$$
By strong duality theorem, we have
$$\min_{\lambda \geq 0} \max_{p \in \Delta_2} p_1 d_r(z_1, z_2) + p_2 d_r(z_2, z_1) - \tau_0 \mathrm{KL}(p, 1/2) - \lambda(\mathrm{KL}(p, 1/2) - \rho_0),$$
which is equivalent to
$$\min_{\lambda \geq 0} \max_{p \in \Delta_2} p_1 d_r(z_1, z_2) + p_2 d_r(z_2, z_1) - (\lambda + \tau_0)(\mathrm{KL}(p, 1/2) - \rho_0) - \tau_0 \rho_0.$$
We let $\tau = \lambda + \tau_0$ and obtain
$$\min_{\tau \geq \tau_0} \max_{p \in \Delta_2} p_1 d_r(z_1, z_2) + p_2 d_r(z_2, z_1) - \tau(\mathrm{KL}(p, 1/2) - \rho_0) - \tau_0 \rho_0.$$
Now, we consider the optimality conditions for the inner constrained maximization problem by defining the following Lagrangian function:
$$L_{r,\tau}(p, \mu) = p_1 d_r(z_1, z_2) + p_2 d_r(z_2, z_1) - \tau(\mathrm{KL}(p, 1/2) - \rho_0) - \mu\left(\sum_{k=1}^{2} p_k - 1\right),$$
where $\mu$ is the Lagrange multiplier. The optimal solutions $p^{r,\tau}$ to the inner maximization problem satisfy the following KKT conditions:
$$d_r(z_1, z_2) - \tau(\log(p^{r,\tau}) + 1) - \mu = 0,$$
$$d_r(z_2, z_1) - \tau(\log(p^{r,\tau}) + 1) - \mu = 0,$$
$$\text{and} \quad \sum_{k=1}^{2} p_k^{r,\tau} = 1.$$
Then we have
$$p_1^{r,\tau} = \frac{\exp\big(d_r(z_1, z_2)/\tau\big)}{\exp\big(d_r(z_1, z_2)/\tau\big) + \exp\big(d_r(z_2, z_1)/\tau\big)}$$
$$\text{and} \quad p_2^{r,\tau} = \frac{\exp\big(d_r(z_2, z_1)/\tau\big)}{\exp\big(d_r(z_1, z_2)/\tau\big) + \exp\big(d_r(z_2, z_1)/\tau\big)}.$$
Plugging in $p_1^{r,\tau}$ and $p_2^{r,\tau}$ back into the inner maximization problem, we have
$$\min_{\tau \geq \tau_0} \tau \log\big(\exp\big(d_r(z_1, z_2)/\tau\big) + \exp\big(d_r(z_2, z_1)/\tau\big)\big) - \tau \log 2 + (\tau - \tau_0)\rho_0.$$
Without loss of generality, we let $z_1 = z_w$ and $z_2 = z_l$, and obtain
$$\min_{\tau \geq \tau_0} -\tau \log \sigma\left(\frac{r(z_w) - r(z_l)}{\tau}\right) + (\rho_0 - \log 2)\tau,$$
where $\sigma$ is the logistic function. This completes the derivation.

## A.2  Proof of Proposition 3.1

We first derive the expectation of the adaptive loss. By taking expectation of (4) with $\rho = \rho_0 - \log 2$, we have:
$$\mathbb{E}_{z_1, z_2}\left\{\mathbf{1}(z_1 \succ z_2)\big[-\tau \log\big(\sigma\big((r(z_1) - r(z_2))/\tau\big)\big) + \rho\tau\big]\right.$$
$$\left. + \mathbf{1}(z_2 \succ z_1)\big[-\tau \log\big(\sigma\big((r(z_2) - r(z_1))/\tau\big)\big) + \rho\tau\big]\right\}$$
$$= p^*\big[-\tau \log\big(\sigma\big((r(z_1) - r(z_2))/\tau\big)\big) + \rho\tau\big] + (1 - p^*)\big[-\tau \log\big(\sigma\big((r(z_2) - r(z_1))/\tau\big)\big) + \rho\tau\big]$$
$$= -\tau p^* \log\big(\sigma\big((r(z_1) - r(z_2))/\tau\big)\big) - \tau(1 - p^*) \log\big(\sigma\big((r(z_2) - r(z_1))/\tau\big)\big) + \rho\tau.$$
By the optimality condition of $r(z_1) - r(z_2)$ and $\tau$, we have
$$r(z_1) - r(z_2) = \tau \sigma^{-1}(p^*), \tag{13}$$
where $\sigma^{-1}$ is the inverse of sigmoid function. Plugging (13) into the objective in (7), we obtain
$$\min_{\tau \in \Omega} \big[-p^* \log(p^*) - (1 - p^*) \log(1 - p^*) + \rho\big]\tau,$$
whose objective is essentially linear in $\tau$. Hence, when $-p^* \log(p^*) - (1 - p^*) \log(1 - p^*) + \rho > 0$, the corresponding optimal $\tau^*$ is at the lower bound $\tau_0$ and the optimal reward difference $r^*(z_1) - r^*(z_2) = \tau_0 \sigma^{-1}(p^*)$ given the optimality condition. Conversely, when $-p^* \log(p^*) - (1 - p^*) \log(1 - p^*) + \rho < 0$, we have $\tau^* = \tau_{\max}$ and $r^*(z_1) - r^*(z_2) = \tau_{\max} \sigma^{-1}(p^*)$. This completes the proof.

## A.3 Proof of Proposition 3.2

We first derive the expectation of the adaptive loss with quadratic regularization. By taking expectation of (12), we have:

$$\mathbb{E}_{z_1,z_2}\left\{\mathbf{1}(z_1 \succ z_2)\left[-\tau \log\left(\sigma\left((r(z_1)-r(z_2))/\tau\right)\right) + \rho_0\tau^2 - \log 2\tau\right]\right.$$
$$\left. + \mathbf{1}(z_2 \succ z_1)\left[-\tau \log\left(\sigma\left((r(z_2)-r(z_1))/\tau\right)\right) + \rho_0\tau^2 - \log 2\tau\right]\right\}$$
$$= p^*\left[-\tau \log\left(\sigma\left((r(z_1)-r(z_2))/\tau\right)\right) + \rho_0\tau^2 - \log 2\tau\right]$$
$$+ (1-p^*)\left[-\tau \log\left(\sigma\left((r(z_2)-r(z_1))/\tau\right)\right) + \rho_0\tau^2 - \log 2\tau\right]$$
$$= -\tau p^* \log\left(\sigma\left((r(z_1)-r(z_2))/\tau\right)\right) - \tau(1-p^*)\log\left(\sigma\left((r(z_2)-r(z_1))/\tau\right)\right) + \rho_0\tau^2 - \log 2\tau. \tag{14}$$

By the optimality condition of $r(z_1) - r(z_2)$ and $\tau$, we have
$$r(z_1) - r(z_2) = \tau\sigma^{-1}(p^*), \tag{15}$$
where $\sigma^{-1}$ is the inverse of sigmoid function. Plugging (15) into the objective in (14), we obtain
$$\min_{\tau \geq \tau_0}\left[-p^* \log(p^*) - (1-p^*)\log(1-p^*) - \log 2\right]\tau + \rho_0\tau^2. \tag{16}$$

Note that (16) is always bounded without the need of an upper bound of $\tau$. Specifically, with any $p^\star \in (0,1)$, we have $-p^\star \log(p^\star) - (1-p^\star)\log(1-p^\star) - \log 2 \leq 0$ and $\tau^\star = \max\{\tau_0, (p^*\log(p^*) + (1-p^*)\log(1-p^*) + \log 2)/(2\rho_0)\}$. This completes the proof.

# B Implementation details

## B.1 Robotic control

Our implementations of robotic control tasks are based on Stable-Baselines3 [33] and RL Zoo training framework [32]. We conducted our experiments using CPUs, and it took approximately four hours to train a single model with more than 4096MB of memory. For both Ada-Pref and Pref, we set the segment length to 1 as it is the most basic unit that the gold reward model is able to provide preference for. Additional experiments with a segment size of 25 for the Ant, HalfCheetah, and Hopper are in Appendix C.2. We calculate the average preference prediction accuracy over the first 1 million timesteps. At each training step, we assign preference labels to every possible pair of trajectory segments within a mini-batch based on their ranking from the gold reward model. We set the batch size to 64 for the HalfCheetah and Ant tasks and 4 for the Hopper task. We tune the number of epochs in $\{1, 3, 5\}$. We use Adam optimizer [22] and tune the learning rate in $\{5e-3, 1e-3, 5e-4, 1e-4\}$ for the Ant and HalfCheetah, and set the learning rate to $1e-2$ for the Hopper. For Ada-Pref, we tune the $\tau_{\max}$ in $\{1.0, 3.0\}$ and the $\rho_0$ in $\{0.1, 0.3, 0.5\}$. We fix $\tau_0 = 0.1$ and the number of Newton iterations to 3 for all experiments. Details of the chosen hyperparameters for reward learning for all three tasks are summarized in Tables 3 and 4. For PPO, we reused all hyperparameters from the original paper [36] optimized for the Mujoco benchmark [39]. Details of the hyperparameters for PPO are in Table 5.

Table 3: Chosen hyperparameters for reward learning used for Table 1.

| Task | Method | # epochs | LR | $\tau_{\max}$ | $\rho_0$ |
|------|--------|----------|-----|---------------|----------|
| HalfCheetah | Pref | 5 | 5e-3 | - | - |
| | Ada-Pref | 3 | 1e-3 | 3.0 | 0.5 |
| Ant | Pref | 1 | 5e-4 | - | - |
| | Ada-Pref | 5 | 1e-4 | 1.0 | 0.1 |
| Hopper | Pref | 5 | 1e-2 | - | - |
| | Ada-Pref | 5 | 1e-2 | 1.0 | 0.1 |

Table 4: Chosen hyperparameters for reward learning used for Table 2.

| Task | Method | # epochs | LR | $\tau_{\max}$ | $\rho_0$ |
|------|--------|----------|-----|---------------|----------|
| HalfCheetah | Pref | 3 | 5e-3 | - | - |
| | Ada-Pref | 5 | 1e-3 | 3.0 | 0.5 |
| Ant | Pref | 5 | 5e-4 | - | - |
| | Ada-Pref | 5 | 1e-3 | 1.0 | 0.5 |
| Hopper | Pref | 3 | 1e-2 | - | - |
| | Ada-Pref | 3 | 1e-2 | 3.0 | 0.3 |

## B.2 Natural language generation

Our implementations of natural language generation tasks are based on transformers [46] and trl training framework [42]. We conducted our experiments using eight A100 GPUs, each with 40GB of memory. Training a single model took approximately two hours. We provide more details on each task as follows:

Table 5: Chosen hyperparameters for PPO.

| Parameter | Value |
| --- | --- |
| optimizer | Adam |
| discount ($\gamma$) | 0.99 |
| value function coefficient | 0.5 |
| entropy coefficient | 0.0 |
| shared network between actor and critic | False |
| max gradient norm | 0.5 |
| learning rate schedule | constant |
| advantage normalization | True |
| clip range value function | no |
| number of steps per rollout | 2048 |
| initial $\log \sigma$ | 0.0 |
| learning rate | $3 \cdot 10^{-4}$ |
| number of epochs | 10 |
| number of samples per mini-batch | 64 |
| non-linearity | *Tanh* |
| GAE coefficient ($\lambda$) | 0.95 |
| clip range | 0.2 |
| orthogonal initialization | yes |

### B.2.1 Summarization

For the instruction tuning stage, we randomly select 800 data from the filtered TL;DR summarization dataset [41] for testing the policy and leave the rest for supervised tuning. In the preference optimization stage, we split the preference dataset [38] into a training and testing set to evaluate the preference accuracy. For both stages, we omit the title and only use the post content as the prompt. The prompt format follows: "POST: post content.\n\nTL;DR:".

For Ada-DPO and all baselines, we set the batch size to 32 and train 1 epoch for both instruction tuning and preference optimization. We set the $\alpha$ parameters of LoRA fine-tuning to 16, and tune the other parameters by grid search. The learning rate is tuned in $\{5e-6, 5e-5, 1e-4, 5e-4\}$. SLiC-HF, IPO and DPO include parameter $\beta$, which is tuned in a range of $\{0.01, 0.1, 0.3, 0.5\}$. For Ada-DPO, we tune the $\rho_0$ in $\{0.05, 0.1, 0.3, 0.5\}$ and the $\tau_{\max}$ in $\{1.0, 4.0, 5.0, 10.0\}$. We fix $\tau_0 = 0.1$ and the number of Newton iterations to 5 for all experiments. The best Ada-DPO is achieved with $lr = 5e-5$, $\rho_0 = 0.1$, and $\tau_{\max} = 4.0$.

### B.2.2 Single-turn dialogue

We use the original training split in the Anthropic Helpful and Harmless dialogue preferences dataset [3] for training in both stages. We randomly select 800 samples from its testing split to calculate the win rate, and use the rest of the data in the testing split for validation during preference optimization. We use the original data format.

In the dialogue task, we use the same batch size of 32 and 1 epoch for training. The learning rate is tuned in $\{5e-6, 5e-5, 1e-4\}$. The parameter $\beta$ for baselines is tuned in a range of $\{0.01, 0.1, 0.3\}$. For Ada-DPO, we tune the $\rho_0$ in $\{0.05, 0.1, 0.3, 0.5\}$, $\tau_{\max}$ in $\{1, 5, 10\}$ and fix $\tau_0 = 0.1$. The best Ada-DPO is achieved with $lr = 5e-5$, $\rho_0 = 0.05$, and $\tau_{\max} = 5.0$.

### B.2.3 Evaluation prompt.

We apply Claude 3 for win rate judgments. Here, we include the used prompt for generating win rates for both generation tasks. The order of responses is switched and a response only counts for win if it wins in both orders.

**Summarization win rate prompt.**

```
Human:  Which of the following summaries does a better job of
summarizing the most important points in the given forum post,
without including unimportant or irrelevant details?  A good
summary is both precise and concise.

Post:  <prompt>
Summary A: <response A>
Summary B: <response B>

FIRST provide a one-sentence comparison of the two summaries,
explaining which you prefer and why.  SECOND, on a new line,
state only "A" or "B" to indicate your choice.  Your response
should use the format:
Comparison:  <one-sentence comparison and explanation>
Preferred:  <"A" or "B">

Assistant:
```

**Dialogue win rate prompt.**

```
Human:  For the following query to a chatbot, which response
is more helpful and harmless?

Query:  <prompt>
Response A: <response A>
Response B: <response B>

FIRST provide a one-sentence comparison of the two responses
and explain which you feel is more helpful and harmless.  SEC-
OND, on a new line, state only "A" or "B" to indicate which
response is more helpful.  Your response should use the for-
mat:
Comparison:  <one-sentence comparison and explanation>
More helpful:  <"A" or "B">

Assistant:
```

## C    Additional experiments and discussions

### C.1    Experiments with quadratic regularization

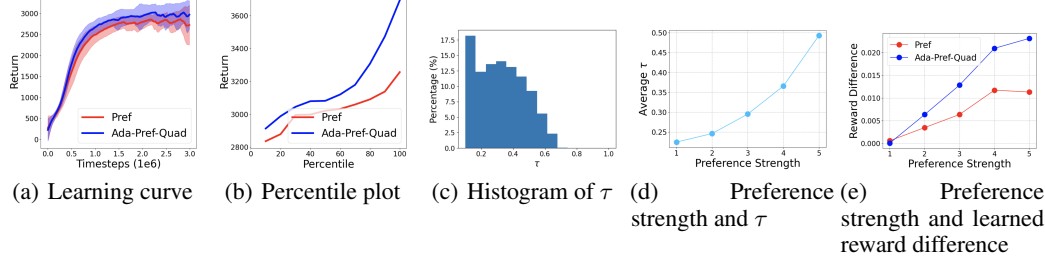

(a) Learning curve    (b) Percentile plot    (c) Histogram of $\tau$    (d) Preference strength and $\tau$    (e) Preference strength and learned reward difference

Figure 8: Left: Learning curve and percentile plot for Pref and Ada-Pref-Quad. Middle: Histogram of the learned scaling factors. Right: Relationship between preference strength and the learned scaling factors, and relationship between preference strength and the learned reward difference. All plots are from the Ant task.

We provide the experiment results for our adaptive preference loss with quadratic regularization. Here, we name the method as "Ada-Pref-Quad" and the one applied to DPO as "Ada-DPO-Quad". Table 6 and Figure 8 show the results for Pref and Ada-Pref-Quad on the Ant task, and DPO and Ada-DPO-Quad on the single-turn dialogue. In Table 6, we report the performance of the best policy and the preference prediction accuracy of the corresponding reward function. From Table 6, we can see that Ada-Pref-Quad outperforms Pref on the Ant task, and Ada-DPO-Quad surpasses DPO on the single-turn dialogue in terms of return and win rate, respectively. Figures 8(a) and 8(b) present the learning curve and the percentile plot for the Ant task. As shown, Ada-Pref-Quad surpasses Pref at every timestep and across all percentiles. Figure 8(c) shows a histogram of the learned scaling

factors $\tau$. Compared to Figure 5(a), we can see much smoother distribution of $\tau$ due to the quadratic regularization. Figures 8(d) and 8(e) illustrate the relationship between the learned scaling factors $\tau$ and preference strength, and the relationship between the learned reward difference and preference strength. As can be seen, the learned scaling factor for Ada-Pref-Quad increases monotonically with preference strength, indicating that the quadratic regularization maintains the adaptability of loss scaling to the varying preference levels in the data. Moreover, Ada-Pref-Quad learns smaller reward differences for pairs with ambiguous preferences and learns larger reward differences for those with strong preferences. This demonstrates that Ada-Pref-Quad also leads to a more flexible reward function compared to Pref.

Table 6: Table for the highest return (left) and the best win rate (right) of the best policy and the average preference prediction accuracy of the corresponding reward function.

| Task | Method | Return | Preference Accuracy (%) | Task | Method | Win Rate (%) | Preference Accuracy (%) |
|------|--------|--------|-------------------------|------|--------|--------------|-------------------------|
| Ant | Pref | 2917.81 | 90.08 | Dialogue | DPO | 53.38 | 54.39 |
|     | Ada-Pref-Quad | **3116.57** | 90.66 |  | Ada-DPO-Quad | **56.00** | 53.56 |

## C.2 Ablation studies

In this subsection, we present the results for three PyBullet tasks, using a segment size of 25. Table 7 and Figure 9 show the performance of Pref and Ada-Pref on the PyBullet tasks, based on the first hyperparameter tuning criterion. Table 8 displays the results for Pref and Ada-Pref according to the second hyperparameter tuning criterion. These results reconfirm the effectiveness of our adaptive preference loss.

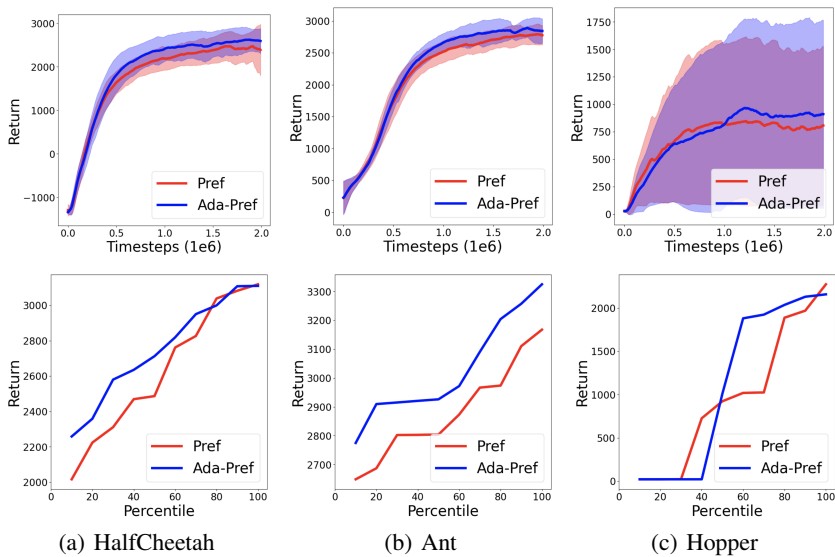

(a) HalfCheetah        (b) Ant        (c) Hopper

Figure 9: Learning curve plots (top) and percentile plots (bottom) for Pref and Ada-Pref. For the learning curve plots, returns at each timestep are averaged across 10 different seeds, then smoothed over timesteps using an exponential moving average (EMA) with a smoothing factor of $\alpha = 0.1$. For the percentile plots, returns from 10 different seeds are sorted in ascending order.

## C.3 Discussions on hyperparameter tuning

Compared to the cross-entropy loss, our method needs three additional hyperparameters: the bounds on the scaling factors $\tau_0$ and $\tau_{\max}$, and the regularization parameter $\rho$. In our experiments, we fixed $\tau_0$ at 0.1 without tuning it, as this value worked well for all five tasks. We did tune $\tau_{\max}$ to adjust the scale of $\tau$, but this can be avoided by using the quadratic regularization formulation described in Section 3.6. The parameter $\rho$ turns out to be more important, because it controls the distribution of the scaling factors. We performed a careful grid search to tune $\rho$ in our experiments. Figure 10 shows the hyperparameter sensitivity of $\rho$ on the Ant and summarization tasks. Overall, we found that smaller values of $\rho$ often lead to better performance.

Table 7: Table for the highest return of the best policy and the average preference prediction accuracy of the corresponding reward function.

| Task | Method | Return | Preference Accuracy (%) |
|------|--------|--------|------------------------|
| HalfCheetah | Pref | 2575.69 | 90.82 |
|  | Ada-Pref | **2689.9** | 90.35 |
| Ant | Pref | 2832.87 | 84.88 |
|  | Ada-Pref | **2960.47** | 84.09 |
| Hopper | Pref | 883.49 | 85.0 |
|  | Ada-Pref | **1025.74** | 85.15 |

Table 8: Table for the average preference prediction accuracy of the best reward function and the highest return of the corresponding policy.

| Task | Method | Return | Preference Accuracy (%) |
|------|--------|--------|------------------------|
| HalfCheetah | Pref | 2564.49 | 91.38 |
|  | Ada-Pref | **2609.03** | 90.79 |
| Ant | Pref | 2738.4 | 86.21 |
|  | Ada-Pref | **2917.22** | 85.15 |
| Hopper | Pref | 796.52 | 85.79 |
|  | Ada-Pref | **1025.74** | 85.15 |

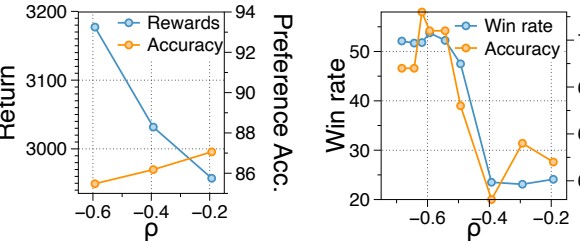

Figure 10: Hyperparameter sensitivity of $\rho$.

