# OpenReview forum: "Adaptive Preference Scaling for Reinforcement Learning with Human Feedback"
_NeurIPS.cc/2024/Conference — NeurIPS 2024 poster_

### Official Review · Reviewer_4UPH · 2024-06-20

**Soundness:** 3
**Presentation:** 2
**Contribution:** 2
**Rating:** 6
**Confidence:** 4

**Summary:**

To learn a versatile rewards essential for the downstream policy optimization, this paper introduces a novel adaptive preference loss function inspired by distributionally robust optimization (DRO).
The proposed approach incorporates an learnable instance-specific scaling factor to accommodate varying uncertainties of preference strength and change the scaling between the preference distribution and reward difference to be non-linear.
The proposed loss function for learning the scaling parameters is strictly convex, and thus they only induce negligible additional computational overhead.
Experiment results on robotic control tasks and LLMs verify the proposed method.

**Strengths:**

1. Adapting the optimization strength by factoring in the characteristics of each preference pair is an interesting direction.
2. The proposed method is well (theoretically) motivated.
3. The proposed method work well in practice.

**Weaknesses:**

1. The adaptive scaling is defined on a per-instance basis, which necessitates significant compute costs and hinders real-world mini-batch learning. This is evident from Appendix, where the algorithm box shows that the training process operates in a per-datapoint manner
2. Some of the derivation may be simplified, both to make the paper more succinct and leave room for algorithm box as well as other technical details.
3. L221: could you theoretically justify the incorporation of adaptive preference scaling into DPO. e.g., in the framework of KL-control as in the original DPO paper?

**Questions:**

1. L71 "Prior work on this topic is very limited.": There are actually a vast pool of literature concerning loss functions for (RLHF's) reward modeling, a quick pointer is [1] for language tasks and [2]  for image generation, as well as the literature review in them. I encourage the authors to add proper discussions and citations to make the claims more precise.
2. L140-141: Could you elaborate this sentence more: "For some applications, it may lead to a reward function that is not flexible enough to differentiate a pair of segments"? In particular, why is the resulted reward function not flexible enough? And how is this related to the linear scaling?
3. In Eq. (3), why do we need to have $KL(p, 1/2)$ in both the objective function and constraint?



***
[1] Yang, Shentao, et al. "Preference-grounded token-level guidance for language model fine-tuning." Advances in Neural Information Processing Systems 36 (2023).

[2] Yang, Shentao, Tianqi Chen, and Mingyuan Zhou. "A Dense Reward View on Aligning Text-to-Image Diffusion with Preference." Forty-first International Conference on Machine Learning. 2024.

**Limitations:**

A discussion on limitations and/or potential negative societal impact seems missing.

---

> ### Author Rebuttal · Authors · 2024-08-07
>
> We would like to thank you for your constructive comments! In the following, your comments are first started and then followed by our point-by-point responses.
>
> **W1: The adaptive scaling is defined on a per-instance basis, which necessitates significant compute costs and hinders real-world mini-batch learning.**
>
> Response to this concern is included in point 3 in the global rebuttal. The additional cost of our proposed method is negligible and it does not hinder mini-batch learning.
>
> **W2: Some of the derivation may be simplified, both to make the paper more succinct and leave room for algorithm box as well as other technical details.**
>
> Thank you for the suggestion. We will simplify some duplicate derivation in section 3 and include the algorithm box and technical details, such as a discussion on additional computational costs, in the main page.
>
> **W3: could you theoretically justify the incorporation of adaptive preference scaling into DPO. e.g., in the framework of KL-control as in the original DPO paper?**
>
> DPO start from the policy optimization objective:
> $$\\max\_\\pi \\mathbb{E}\_{s,a\\sim\\pi (a|s)}[r(s,a)] - \\beta D\_{\\mathrm{KL}}[\\pi(a|s), \\pi\_{\\mathrm{ref}(a|s)}]$$
> and make use of the optimality condition given that the horizon is 1:
> $\pi_r(a|s) = \frac{1}{Z(s)} \pi_{\mathrm{ref}}(a|s) \exp \left( \frac{1}{\beta}r(s,a) \right).$
> Then they reparameterizes the reward function as $r(s,a)=\beta\log(\pi_r(a|s)/\pi_{\mathrm{ref}}(a|s))+\beta\log Z(s)$ and employs the standard cross-entropy loss for reward learning.
> Our proposed Ada-DPO retains the same reparametrization and only modifies the reward learning loss to Equation (6). The derivation and analysis for this change in the reward loss are included in Sections 3.2 and 3.3.
>
> We assume the reviewer is suggesting moving the adaptive scaling factors to the KL-constrained policy optimization objective. Unfortunately, we found this difficult due to the regularization on scaling factor. However, at a high level, as mentioned in Line 222, we can merge the regularization parameter $\beta$, which controls the KL term, with the scaling factors in the final objective. Thus, our method can be viewed as adapting a different KL-control term for each preference pair. For strong preference data, we learn a large scaling factor, corresponding to a smaller KL-control term, allowing the model to deviate more from the reference policy. In contrast, a larger KL-control term is used for ambiguous preference data.
>
> **Q1: "Prior work on this topic is very limited.": There are actually a vast pool of literature concerning loss functions for (RLHF's) reward modeling, a quick pointer is [1] for language tasks and [2] for image generation, as well as the literature review in them. I encourage the authors to add proper discussions and citations to make the claims more precise.**
>
> Thank you for pointing out the related works. We will include discussions and citations of these references in our next version. After carefully reviewing the mentioned works, we find that both focus on changing the reward function, but the used loss is still based on cross-entropy. This makes them complementary to our work, as we are improving the loss function. Specifically:
>
> [1] introduces a token-level preference reward learning loss instead of the standard sequence-level objective, addressing the granularity mismatch between preferences and LM training losses. Their approach employs a more fine-grained reward function. However, the cross-entropy loss is still employed when there are two responses, which means their method can potentially be combined with ours.
>
> [2] focuses on the preference alignment of text-to-image diffusion models and proposes temporal discounting DPO-style objectives that consider the sequential nature of the generation process. However, their method is specific to diffusion models and cannot be applied to general RLHF as ours can. Additionally, the used loss is still based on cross-entropy.
>
> In summary, we consider these two works to be orthogonal and complementary to our work.
>
>
> **Q2: Could you elaborate this sentence more: "For some applications, it may lead to..."**
>
> Response to this concern is included in point 2 in the global rebuttal.
>
> **Q3: In Eq. (3), why do we need to have $\mathrm{KL}(p,1/2)$ in both the objective function and constraint?**
>
> Including the KL term in both the objective and the constraints is a common practice in constrained distributionally robust optimization (DRO) problems, as shown in Eq. (1) in [3]. Without the KL constraints, the loss becomes the cross entropy loss with temperature $\tau_0$, making the loss not be able to adapt to each pair of samples. Without the KL term in objective, the objective will not be smooth and hard to optimize.
>
>
> **L1: A discussion on limitations and/or potential negative societal impact seems missing.**
>
>  Response to this concern is included in point 4 in the global rebuttal.
>
> ### References
>
> [1] Yang, Shentao, et al. 'Preference-grounded token-level guidance for language model fine-tuning.' Advances in Neural Information Processing Systems 36 (2023).
>
> [2] Yang, Shentao, Tianqi Chen, and Mingyuan Zhou. 'A Dense Reward View on Aligning Text-to-Image Diffusion with Preference.' Forty-first International Conference on Machine Learning. 2024.
>
> [3] Qi Qi, Jiameng Lyu, Kung sik Chan, Er Wei Bai, Tianbao Yang. Stochastic Constrained DRO with a Complexity Independent of Sample Size. Transactions on Machine Learning Research, 2023.

---

> > ### Comment · Reviewer_4UPH · 2024-08-10
> > **Response to the authors**
> >
> > Dear authors,
> >
> > Thank you so much for the helpful responses. I hope these discussions and revisions can be incorporated into the next version of the manuscript. I've increased my rating to 6.

---

> > > ### Author Response · Authors · 2024-08-11
> > > **Thank you for the thorough review**
> > >
> > > Dear Reviewer,
> > >
> > > Thank you for your thoughtful feedback and for increasing the scores toward acceptance. We will incorporate your suggestions in our next version.
> > >
> > > Best regards,
> > >
> > > The Authors

---

### Official Review · Reviewer_ZDuk · 2024-07-12

**Soundness:** 3
**Presentation:** 3
**Contribution:** 3
**Rating:** 5
**Confidence:** 4

**Summary:**

This paper studies the problem of learning from preference data and introduces a learnable scaling parameter for each preference sample. The authors propose an adaptive preference loss function that assign small scaling parameters to ambiguous preferences pairs and large scaling parameters to clear preferences. Experiments demonstrate the improvement of policy optimization performance and efficient of the hyperparameter tuning.

**Strengths:**

Considering different samples have different preference strength, using an adaptive preference scaling makes sense for preference learning.

Experiments are conducted on both robotic control and natural language generation tasks.

**Weaknesses:**

The increase in flexibility of the proposed reward function is not verified. The authors claim that one of the limitations of the BT model is the logit of the preference distribution scales linearly with the reward difference. However, according to Proposition 3.1, the limitation still exists. There is no empirical result that supports the claim.

Missing the ablation study of the regularization term in the proposed loss function.

Missing the RLHF baseline in the natural language generation task.

“Ada-Pref demonstrates greater resistance to performance degradation than Pref, indicating its superior ability to align the learned reward function with policy optimization.” It seems that it is not true in the Hopper task?

Minors:
Line 120: The notation of state transition function is conflicted with preference distribution.
Please clarify the meaning of M and K in Algorithm 1.

**Questions:**

I do not believe the proposed loss function is convex, can you provide the proof of Remark 3.1?
Why the reward models trained with the proposed method can better guide policy model selection considering the reward model performance is not improved?

How do the authors obtain the true reward difference in Section 4.3?

**Limitations:**

The computation overhead introduced by learning the scaling parameter is not discussed.

---

> ### Author Rebuttal · Authors · 2024-08-07
>
> Thank you for your thoughtful comments! We sincerely appreciate your time in reading the paper, and our point-to-point responses to your comments are given below.
>
> **W1: The increase in flexibility of the proposed reward function is not verified. The authors claim that one of the limitations of the BT model is the logit of the preference distribution scales linearly with the reward difference. However, according to Proposition 3.1, the limitation still exists. There is no empirical result that supports the claim.**
>
> Thank you for the suggestion. We provide additional clarification to better demonstrate the increased flexibility of the proposed method:
> * We would like to clarify that the proposed method no longer maintains a restrictive linear relationship between the logit of the preference distribution and the reward difference. In Proposition 3.1, we show that the relationship between the optimal reward difference and the logit of the true preference distribution varies for strong preference pairs and ambiguous pairs, resulting in a more flexible non-linear relationship. Additionally, during training, the learned logit is often not optimal, meaning the scaling factor is not strictly on the bounds. This is illustrated in Figures 5a and 6, where the scaling factor in the middle can create a more complex correspondence. In contrast, the BT model always maintains the same linear relationship between the learned logit and the learned reward difference during training, restricting flexibility.
>
> * Directly measuring the flexibility of the reward function empirically can be challenging since it's difficult to quantify. To better support our claim, we provide some supporting empirical evidence in the paper. For instance:
>     * Figure 5(c) shows that Ada-Pref learns smaller reward differences for pairs with ambiguous preferences and larger reward differences for pairs with strong preferences, indicating a wider range of learned reward differences.
>     * Figure 7 provides examples showing that Ada-DPO learns larger reward differences than DPO for clear pairs and smaller reward differences than DPO for ambiguous pairs. We also include more examples in the response to the second reviewer.
>
> **W2: Missing the ablation study of the regularization term in the proposed loss function.**
>
> The ablation study of the regularization term $\rho$ in the proposed loss function has been provided in Appendix D.3 due to space constraints. If we remove the regularization term, the proposed loss collapses to the standard DPO loss with a modified $\beta' = \beta \cdot \tau_0$, which performs significantly worse than the reported DPO results. Consequently, we did not include it in the main text. For reference, removing the regularization term achieves a win rate of 24.56 on the summarization task.
>
> **W3: Missing the RLHF baseline in the natural language generation task.**
>
> Thank you for pointing out the absence of an RLHF baseline in the natural language generation task. We understand the importance of experimenting with PPO instead of DPO to fully comprehend the impact of our method on RM's signal. Due to resource constraints, we prioritized experiments requiring PPO for robotic control tasks. Conducting PPO experiments for natural language generation tasks presents significant computational challenges, including the need to store and manage both reward and critique models (which are LLMs). PPO also involves more training steps and numerous hyperparameter tunings, making it difficult to implement within a resource-limited environment. Therefore, we opted for DPO as an alternative in our submission.
>
> Due to the limited resources and the need to run other experiments, we are currently unable to provide PPO results for natural language generation tasks. Nonetheless, we will try our best to update the results before the discussion deadline.
>
> **W4: It seems that Ada-Pref's greater resistance to performance degradation is not true in the Hopper task?**
>
> This is true in the Hopper task as well. Comparing Tables 1 and 2, we can see that Ada-Pref drops by 32.9\% in the Hopper task, while Pref drops by 43.7\%.
>
> **Q1-A: I do not believe the proposed loss function is convex, can you provide the proof of Remark 3.1?**
>
> The proposed loss function is strictly convex with respect to each $\tau_i$ when $r(z_w) \neq r(z_l)$. For simplicity, we omit the index $i$ from $\tau_i$ and set $C = r(z_w) - r(z_l)$. Then the instance-level loss function is written as:
> $$f(C, \tau) = -\tau \log \sigma(C/\tau) + \rho\tau. $$
> Since $\frac{\partial f}{\partial \tau} = \frac{C^2}{\tau^3} \sigma'(C/\tau) > 0$ for all $\tau \in [\tau_0, \tau_{\mathrm{max}}]$, $f$ is strictly convex with respect to $\tau$.
>
> **Q1-B: Why the reward models trained with the proposed method can better guide policy model selection considering the reward model performance is not improved?**
>
> We cannot directly evaluate the performance of the reward model solely based on preference prediction accuracy. This is because preference prediction accuracy does not fully capture the effectiveness of the reward model in the context of policy optimization. It only reflect the sign of reward difference but not the scale. Our reward function is designed to be more flexible, providing distinct rewards for clear pairs and comparable rewards for ambiguous pairs. This flexibility allows the reward function to generate a wider range of rewards, which is crucial for effective downstream policy optimization.
>
>
> **Q2: How do the authors obtain the true reward difference in Section 4.3?**
>
> PyBullet environments provide the ground truth reward for each pair of state and action, which are then used to obtain the true reward difference in Section 4.3.
>
> **Q3: The computation overhead introduced by learning the scaling parameter is not discussed.**
> Response to this concern is included in point 3 in the global rebuttal.

---

> > ### Comment · Reviewer_ZDuk · 2024-08-12
> > **Response to the authors**
> >
> > Thank you for the informative response!
> >
> > Can you further visualize situations in which the BT model cannot express, maybe in your synthetic setting where the true preference strengths can be captured? The learned scaling parameter also seems linear regarding preference strength, as shown in Figure 5(b).
> >
> > If this motivation is not well-established, adding Adaptive Preference Scaling that has complicated implementation is not worth the cost. I also wonder if using simpler approaches, such as adding a reward margin to ranking loss or adding another layer of (non-) linear function outside the sigmoid function, can achieve similar or better performance.

---

> > > ### Author Response · Authors · 2024-08-13
> > > **Thank you for the discussion**
> > >
> > > Thank you for your thoughtful comments!
> > >
> > > Due to NeurIPS regulations, we are unable to upload figures or provide links. However, we would like to clarify the following points:
> > >
> > > The BT model assumes a linear relationship between the reward difference and the logit of the preference distribution. This means it cannot accurately represent situations where the relationship is non-linear. In contrast, our method models this relationship as non-linear. Specifically, with linear regularization, the relationship becomes piecewise-linear (see Proposition 3.1), and with quadratic regularization, the relationship is more complexly non-linear (see Proposition 3.2).
> > >
> > > In both cases, it is important to note that when the logit value is small, our method learns a smaller reward difference. Conversely, when the logit value is large, our method learns a larger reward difference compared to the BT model. This adaptive, non-linear relationship makes our method more flexible and better suited for capturing complex preference dynamics that the BT model cannot handle.
> > >
> > > Further examples of the benefits of this non-linear relationship are provided in our response to Q1 of reviewer E9fU. In Example 1 (low preference strength), Example 2 (moderate preference strength), and Example 3 (large preference strength), the reward differences for DPO are 0.64, 1.07, and 1.47, respectively. Meanwhile, the reward differences for Ada-DPO are 0.31, 0.88, and 2.83, respectively, demonstrating that Ada-DPO scales more appropriately across varying levels of preference strength.
> > >
> > > Regarding the learned scaling parameter shown in Figure 5(b), we want to clarify that the near-linearity of the scaling factor does not imply a linear relationship between the learned logits and preference strength.
> > >
> > > Lastly, in the additional experiment, we implemented the reward margin method from [1] on three robotic control tasks. We did not include other reward margin methods, such as those from [2] and [3], because they require additional data called "score", which is expensive to obtain, making a fair comparison difficult. As shown in the table below, the reward margin method from [1] not only fails to match our method’s performance but also performs significantly worse than standard RLHF.
> > >
> > > | Task         | Method   | Return  |
> > > |--------------|----------|---------|
> > > | HalfCheetah  | Pref     | 2724.42 |
> > > |                       | Margin | 577.98 |
> > > |              | Ada-Pref | **2875.45** |
> > > | Ant          | Pref     | 2917.81 |
> > > |              | Margin | 866.5 |
> > > |              | Ada-Pref | **3177.11** |
> > > | Hopper       | Pref     | 1324.91 |
> > > |              | Margin | 40.13 |
> > > |              | Ada-Pref | **1692.1**  |
> > >
> > > [1] Qin, Bowen, Duanyu Feng, and Xi Yang. "Towards Understanding the Influence of Reward Margin on Preference Model Performance." arXiv preprint arXiv:2404.04932 (2024).
> > >
> > > [2] Touvron, Hugo, Louis Martin, Kevin Stone, Peter Albert, Amjad Almahairi, Yasmine Babaei, Nikolay Bashlykov et al. "Llama 2: Open foundation and fine-tuned chat models." arXiv preprint arXiv:2307.09288 (2023).
> > >
> > > [3] Amini, Afra, Tim Vieira, and Ryan Cotterell. "Direct preference optimization with an offset." arXiv preprint arXiv:2402.10571 (2024).

---

> ### Comment · Reviewer_ZDuk · 2024-08-13
>
> I appreciate the authors' detailed reply. They have addressed most of my concerns.
>
> However, there is no evidence (even without a toy example) demonstrating that the BT model would fail due to its linear relationship. This makes it unclear for researchers to determine when the proposed method, which seems hard to re-implement,  should be used.
>
> Therefore, I would keep my previous evaluation.

---

> > ### Author Response · Authors · 2024-08-13
> >
> > Thank you for your insightful comment. We would like to clarify that the BT model does not completely fail but is indeed insufficient in some cases. For example, when the relationship between the ground truth reward difference and the logit is non-linear, the BT model cannot fully capture and learn this correspondence. In our synthetic case, we only know the ground truth reward, not the logit of the preference distribution, so we can't provide direct evidence even in a toy example. However, the superior performance of our method across various tasks suggests that non-linear relationships do exist in real data. While the overall performance of the BT model is ok, our method offers greater flexibility and consistently delivers better results.

---

### Official Review · Reviewer_E9fU · 2024-07-13

**Soundness:** 3
**Presentation:** 3
**Contribution:** 3
**Rating:** 6
**Confidence:** 2

**Summary:**

The paper identifies a limitation in RLHF methods, noting that ranking over pairs of trajectory segments often fails to capture the varying strengths of preferences across different pairs. To address this, the paper proposes a new adaptive preference loss (Ada-DPO), underpinned by distributionally robust optimization (DRO). The proposed method improves policy performance and is supported by theoretical proofs on convexity and univariate analysis.

**Strengths:**

- The paper identifies a key limitation in previous RLHF methods and provides a clear theoretical analysis of its methods and claims.
- It presents robust experiments and quantitative analysis on both robotic controls and natural language generation tasks.

**Weaknesses:**

- It seems that only a single run was conducted for the experiments in the NLP tasks. More repeated runs would be more convincing, especially since the differences in performance are so close. Additionally, p-value testing for that domain would help determine if the differences are truly significant.

**Questions:**

- I would like to see more examples like Figure 7, comparing the learned reward differences with different scaling factors.

**Limitations:**

- More discussion on the limitations of the proposed method and future directions can be added.

---

> ### Author Rebuttal · Authors · 2024-08-07
>
> We would like to thank you for appreciating the feature of the proposed method and are grateful for the constructive comments! In the following, your comments are first stated and then followed by our point-by-point responses.
>
> **W1: It seems that only a single run was conducted for the experiments in the NLP tasks. More repeated runs would be more convincing.**
>
> Thank you for pointing that out. We have included additional results and analysis for more runs in point 1 of the global rebuttal. We can observe from the table that our method consistently achieves stable and significant improvements over the baselines in both tasks. Given the large margin of improvement, we believe p-value is not necessary.
>
> **Q1: I would like to see more examples like Figure 7, comparing the learned reward differences with different scaling factors.**
>
>  We provide three examples with different learned scaling factors in the summarization task below. We do not include the complete prompt here for better readability and space limit.
>
> ----
>
> ### Example 1:
>
> * **Original text**: I need some advice. I've been talking with this girl for about 2 weeks now. We went out last weekend and it went great. We were working on setting up another date and she told me that she was concerned about distance ...
>
> * **Chosen summary**:  agreed to meetup for coffee but haven't heard from her since tuesday night. Want to know what i can do to make it happen again.
>
> * **Rejected summary**:  agreed to meetup but haven't heard from since. Tried texting a couple times just trying to understand whats going on.
>
> This sample pair gets $\tau=0.325$ and the learned reward difference is 0.3071(Ada-DPO) vs 0.6419 (DPO). We can observe that the two responses in this example is similar, with the chosen summary being slightly more specific. Our method learn a small scaling factor and smaller reward difference since the gap between two responses is not that significant.
>
> ----
> ### Example 2:
>
> * **Original text**: I'm not sure if I can even do anything, and if the person in question wasn't an ...
>
> * **Chosen summary**:  store franchise owner is probably stealing from their store by under ordering groceries/charging less than customer pays. possibly unethical behavior by owner of store? not sure how to proceed/act. help pls reddit.
>
> * **Rejected summary**:  store franchise owner probably is hiding their grocery bill from the rest of the store staff and is getting some kind of unethical benefit out of it, not sure if I can do anything. Advice?
>
> This sample pair gets $\tau=0.93$ and the learned reward difference is 0.8786(Ada-DPO) vs 1.066 (DPO). This example present a reasonable difference, with Chosen being more correct and better summarize the text. Our method and the DPO baseline get a similar reward difference and the scaling factor is around 1.
>
> -----
> ### Example 3:
> * **Original text**: So my dream is do stand up comedy, improv comedy, writing and/or sketch comedy full time...
>
> * **Chosen summary**:  I want to pursue stand up comedy full time but I am afraid of losing my brothers rent money and my family. Do I follow my dreams or play it safe? Any advice/criticism is greatly appreciated!
>
> * **Rejected summary**:  Follow your dreams or play it safe?
>
> This sample pair gets $\tau=3.00$ and the learned reward difference is 2.832(Ada-DPO) vs 1.467 (DPO). Our method learn a large scaling factor and a larger reward difference for the last sample. As we can see from the two responses, the preference is obvious, with the rejected summary do not contain important information.
>
> ------
>
>
> **Q2: More discussion on the limitations of the proposed method and future directions can be added.**
>
> Response to this concern is included in point 4 of the global rebuttal.

---

### Official Review · Reviewer_RUFd · 2024-07-15

**Soundness:** 3
**Presentation:** 2
**Contribution:** 2
**Rating:** 5
**Confidence:** 4

**Summary:**

The paper focuses on redesigning the loss function with adaptive scaling parameters to deal with the uncertainty in the preferences and thus improving the reward modeling flexibility.  In the context of both robotics and NLP, the algorithm with the new loss shows improved performance.

**Strengths:**

1. The proposed loss is flexible and can be incorporated in majority of the current RLHF frameworks
2. The objective has strong connections to DRO which is critical since preferences will be noisy and sub-optimal in practical scenarios.

**Weaknesses:**

1. The experimental ablation is weak and doesn't provide concrete indications of the strength of the proposed algorithm. For ex in Figure 3, the win-rate gap is marginal and not significant. Why is the performance in summarization better in Figure 4 is not extremely clear?

2. The experimental evaluations lack important benchmarks and comparisons. For the Robotics environment, it's crucial to compare with Pebble, SURF, and PARL to understand the true efficacy of the proposed approach.

**Questions:**

1. Can the authors provide more concrete mathematical justifications regarding the issue with linear depedence with the logit of the preference distribution? Specifically, connecting with the relation to noisy preferences?
2. "For some applications, it may lead to a reward function that is not flexible enough to differentiate a
141 pair of segments, which are supposed to have significantly different rewards." Can the authors please highlight such examples intuitively or mathematically, why such a collapse might happen?
3. The authors mention "Our proposed method is inspired by DRO, it serves a distinct purpose: improving reward learning in RLHF, which is orthogonal to distributional robustness". Can the authors pls justify this statement and discuss why it's orthogonal? DRO in reward learning will also improve the robustness and flexibility of the reward model to noisy preferences, in what sense it does better than DRO?
4. Can the authors provide a more concrete definition of the best win rate and best prediction accuracy?
5. It will be helpful if the authors can provide more details on how the trajectories are constructed for the robotics experiments.

**Limitations:**

Check above

---

> ### Author Rebuttal · Authors · 2024-08-07
>
> We are grateful for the valuable feedback that you have provided! Please see our response per each of your concerns below:
>
> **W1: Weak experimental ablation. Why is the performance in summarization better in Figure 4 is not extremely clear?**
>
> The response to weak experimental ablation is included in the global rebuttal. In Figure 4, we select the best model based on the preference prediction accuracy and the improvement indicates that our method better aligns the learned reward function and policy optimization, while DPO baseline results in a model with high preference accuracy but low win rate. The gap may be due to that our Ada-DPO is more robust to overfitting, and that the learned reward is more flexible and better guide the policy optimization.
>
> **W2: The experimental evaluations lack important benchmarks and comparisons. For the Robotics environment, it's crucial to compare with Pebble, SURF, and PARL to understand the true efficacy of the proposed approach.**
>
> The proposed method is orthogonal to PEBBLE [1], SURF [2], or PARL [3] due to the distinct focus of our work. Specifically:
>
> * **PEBBLE** aims to enhance both sample and feedback efficiency by integrating unsupervised pre-training with off-policy RL.
> * **SURF** focuses on improving feedback efficiency by utilizing unlabeled data for reward learning.
> * **PARL** tackles the issue of the alignment objective's dependence on data generated by the optimal policy by introducing a bilevel formulation.
>
> In contrast, our method focuses on enhancing reward model training by considering the varying preference strength between each sample pair. This approach is distinct from the goals of improving feedback efficiency or resolving the entanglement between the alignment objective and the trajectory collecting policy.
>
> Our proposed loss function can be readily adapted to enhance the performance of these methods (PEBBLE, SURF, and PARL), serving as a complementary approach rather than a direct comparison. As discussed in Remark 3.2 of the paper, Eq. (4) in PEBBLE [1], Eq. (3) in SURF [2], and Eq. (5) in PARL [3] show that these frameworks still optimize the standard cross-entropy loss for reward learning. Our proposed adaptive loss function can replace this standard cross-entropy loss to potentially enhance their performance.
>
> **Q1: Can the authors provide more concrete mathematical justifications regarding the issue with linear dependence with the logit of the preference distribution? Specifically, connecting with the relation to noisy preferences?**
>
> We want to clarify that the issue of the linear dependence of the BT model is not directly related to noisy preferences. Rather, it limits the flexibility of the learned reward difference, preventing the reward function from providing a wider range of rewards in downstream policy optimization. Our proposed method incorporates scaling factors to achieve a non-linear relationship, allowing the learned reward difference to be more flexible.
>
> We provide a mathematical analysis in Section 3, demonstrating that our method increases the complexity of the reward model compared to the baseline. Specifically, we show that our method learns a smaller reward difference for ambiguous preference data and a larger reward difference for strong preference data. The experimental results further prove that this improved flexibility translates to better performance. We believe this sufficiently demonstrates the linear dependence is not flexible enough and hence can be suboptimal.
>
> **Q2**: Included in global rebuttal.
>
> **Q3: Can the authors pls justify the statement and discuss why the proposed method is orthogonal to DRO?**
>
> While our proposed method borrows the technical concept from DRO, the aims are fundamentally different. DRO focuses on improving robustness against data distribution shifts between training and test sets, which does not necessarily translate to increased flexibility of the reward model that we want to address. In contrast, our method aims to improve flexibility by incorporating scaling parameters to help reward learning, but not targeting robustness against data distribution or noisy preferences. This distinct focus is why we consider our method orthogonal to DRO. More technical differences are discussed in Line 108.
>
> **Q4: Can the authors provide a more concrete definition of the best win rate and best prediction accuracy?**
>
> We use two different criteria to select the model among all those trained with different hyperparameter configurations:
>
> * Best Win Rate: We identify the best policy based on its end performance, specifically the policy with the highest win rate, and then report the win rate and the preference prediction accuracy of the corresponding reward function (Figure 3).
> * Best Prediction Accuracy: We identify the best policy as the one whose corresponding reward function achieves the highest preference prediction accuracy (Figure 4). We present this results because win rate evaluation involves costly proprietary models and tuning based on the preference accuracy could be much more efficient in practice.
>
> **Q5: It will be helpful if the authors can provide more details on how the trajectories are constructed for the robotics experiments.**
>
> Thank you for pointing that out. Due to the wide variety of experiments and the numerous details about the experimental settings, we were unable to include all the detailed explanations in the main text. However, a brief explanation of how the trajectories are constructed for the robotics experiments is provided in Appendix C.1.
>
> ### Reference
> [1] Pebble: Feedback-efficient interactive reinforcement learning via relabeling experience and unsupervised pre-training.
>
> [2] SURF: Semi-supervised reward learning with data augmentation for feedback-efficient preference-based reinforcement learning.
>
> [3] PARL: A unified framework for policy alignment in reinforcement learning

---

> > ### Comment · Reviewer_RUFd · 2024-08-12
> > **Response to Rebuttal by Authors**
> >
> > Thanks for providing concrete answers to my queries and issues. Most of my concerns have been resolved, hence I am increasing my score. Please add the references mentioned in detail and explain the reason for not comparing them with the baselines.
> >
> > However,  mathematical justifications regarding the issue with linear dependence with the logit of the preference distribution is not very clear and would request authors to provide a clear explaination mathematically.

---

> > > ### Author Response · Authors · 2024-08-13
> > > **Thank you for the discussion**
> > >
> > > Thank you for your valuable comments. We will include the mentioned references and the discussion in the paper. Additionally, we would like to address your remaining concern regarding the mathematical justification related to the issue of linear dependence.
> > >
> > > The BT model assumes a linear relationship between the reward difference ($\Delta r$) and the logit of the preference distribution ($\ell$), expressed as $\Delta r = \ell$. This assumption inherently limits the model’s ability to accurately capture scenarios where this relationship is non-linear.
> > >
> > > In contrast, our method models this relationship as non-linear. Specifically, in the case of linear regularization, the relationship is piecewise-linear and can be expressed as: $\Delta r = \tau_0\ell , \text{if } -t < \ell \leq t; \Delta r=\tau_\max\ell, \text{otherwise},$ where $t$ is a threshold related to $\rho$, and $\tau_0 < 1 < \tau_\max$ (see Proposition 3.1). For quadratic regularization, the relationship becomes even more complexly non-linear (see Proposition 3.2). Importantly, in both cases, when $\ell$ is small, $\Delta r$ is smaller, and when $\ell$ is large, $\Delta r$ is larger compared to the BT model. This adaptive, non-linear relationship enhances our method’s flexibility and allows it to better capture complex preference dynamics that the BT model cannot.
> > >
> > > Although we intended to include visualizations of these relationships to make them more intuitive, NeurIPS regulations prevent us from doing so in the rebuttal. However, we will add these plots in the paper to provide better clarification.
> > >
> > > Additionally, please refer to the examples on the summarization task in our response to Q1 for reviewer E9fU, which further illustrate how this non-linearity adapts to the data.
> > >
> > > Lastly, we noticed that, despite your mention of potentially increasing the score, it has remained unchanged. We would greatly appreciate it if you could reconsider the score in light of the explanations we provided.

---

### Author Rebuttal · Authors · 2024-08-07

We would like to thank all the reviewers for the valuable feedback! Before we answer to each of the reviewers individually, we list and address common concerns below:

>  **1. The experimental ablation doesn't provide concrete indications of the strength of the proposed algorithm. Only a single run was conducted for the experiments in the NLP tasks. More repeated runs would be more convincing.**

We would like to highlight that the proposed method achieves non-marginal improvement across multiple tasks. It outperforms DPO baseline by 9-25% percent in terms of return on Ant and Hopper task (Table 1), and achieve 11% higher win rate than DPO on Dialogue task (Figure 3). While the improvement on HalfCheetah and summarization is not as large, they demonstrate that our method is consistently better and effective.

Furthermore, the smaller improvement on summarization could be due to Claude 2 not being a strong enough judge for evaluation. We conduct new experiments in NLP tasks replacing Claude 2 with Claude 3, a stronger judge, and use three different seeds. The results are presented in Table 1 in the attached file, showing that our method significantly improves over DPO and is stable across seeds.

> **2. Clarification on sentence: "For some applications, it may lead to a reward function that is not flexible enough to differentiate a pair of segments...”**

Human preferences are often influenced by numerous factors that interact in non-linear ways, making the BT model suboptimal as a reward model. For example, when the reward difference is small, even slight changes in certain features might lead to significant shifts in preference. The BT model may struggle to capture such rapid shifts due to its slower transition. Conversely, when the reward difference is already large, its effect on the preference distribution could be marginal, meaning that a small increase in preference probability corresponds to a large increase in the reward difference. The BT model, with its linear correspondence, will not learn a large enough reward difference for strong preference data due to its small gradient. Our method increases the gradient (as shown in Figure 1) in this case to more flexibly learn a wider range of rewards. We would like to emphasize that the scale of the reward difference is crucial, as it affects policy optimization, and a larger or smaller reward difference can benefit the learning of some downstream tasks.

> **3. The compute costs of the algorithm is not discussed and could be significant. It hinders real-world mini-batch learning.**

We would like to clarify that the additional cost of our proposed method is negligible, and it does not hinder mini-batch learning. Computationally, since the proposed loss is strictly **convex** and **univariate** in the scaling factor, the inner minimization problem (Lines 5-7 in Alg. 1) can be solved efficiently with a few iterations (k=5) of the Newton method. The additional computational cost of each update is **negligible** compared to the overall RLHF pipeline. Memory-wise, we need to temporarily store $\tau_i$ for each sample, which is minor considering the high-dimensional nature of the data, and we can free the memory after the backward update.

Regarding the mini-batch learning, although the algorithm is presented in a per-data manner to show how each $\tau_i$ is optimized, it can be directly adapted for mini-batch learning. In practice, we update all $\tau_i$ in the mini-batch in parallel. Our experiments also utilize mini-batch learning to ensure efficiency.


> **4. More discussion on the limitations of the proposed method, future directions and societal impact can be added.**

We provide our discussion below and will include these points in our next version: One main limitation of our proposed method is the introduction of three hyperparameters to control the scaling ($\tau_0, \tau_{\max}, \rho$), which induces additional tuning costs. In the paper, we propose an extension to the quadratic penalty in Section 3.6 to eliminate $\tau_{\max}$, but the performance of this extension is not as good. Developing more efficient tuning approaches or regularization techniques to reduce the need for these hyperparameters will be considered as future work. Additionally, extending our adaptive loss to handle ranking data with more than two responses for preference optimization is another potential future direction.

For societal impact, our method can be used to better align the LLM to user preferences, presenting some societal risks. Primarily, same as DPO, it may reinforce and amplify existing biases if the preferences and feedback used in training are skewed or prejudiced. Ethical concerns arise when the model aligns with harmful or unethical preferences.

---

### Decision · Program_Chairs · 2024-09-25

**Decision:**

Accept (poster)

**Comment:**

The paper presents a novel adaptive preference loss for RLHF, integrating a distributionally robust optimization approach to handle varying strengths in human preferences. The reviewers generally appreciate the motivation behind the method and acknowledge its potential impact, especially in aligning rewards with policy optimization across robotic control and NLP tasks.

**Strengths:**
- The proposed method is theoretically sound, with robust experimentation across different domains.
- The flexibility introduced by the adaptive scaling parameters addresses a significant limitation in previous RLHF methods.

**Weaknesses:**
- Reviewers noted concerns about the weak experimental ablation, particularly in NLP tasks, where only a single run was initially conducted. The authors provided additional results with repeated runs, which addressed most concerns, but some reviewers remain unconvinced about the conclusiveness of these results.
- The lack of comparison with key benchmarks like PEBBLE, SURF, and PARL in the robotics domain was pointed out. The authors argue that their method is orthogonal to these benchmarks and could complement rather than directly compete with them.
- The computational overhead of the proposed method was discussed, with the authors asserting that the additional costs are negligible. However, the practical implications of this overhead in real-world settings remain a point of contention among reviewers.

**Unresolved Points:**
- Some reviewers were not entirely satisfied with the mathematical justifications provided regarding the linear dependence issue in preference modeling, indicating that the explanation could be clearer.
- The necessity of the proposed method versus simpler alternatives (e.g., reward margin methods) is still debated, with some reviewers questioning if the complexity of the method is justified by the performance gains.
- Concerns about the practical implementation and scalability of the method in real-world mini-batch learning scenarios were raised and only partially addressed by the authors.

**Recommendation:**
The paper is borderline, with a slight leaning towards acceptance due to its innovative approach and potential impact. However, the unresolved issues, particularly regarding experimental robustness and practical scalability, suggest that further revisions and clarifications are necessary for a more confident endorsement.